# MatSciBench: Benchmarking the Reasoning Ability of Large Language Models in Materials Science

## Abstract

Large Language Models (LLMs) have demonstrated remarkable abilities in scientific reasoning, yet their reasoning capabilities in materials science remain underexplored. To fill this gap, we introduce **MatSciBench**, a comprehensive college-level benchmark comprising 1,340 problems that span the essential subdisciplines of materials science. **MatSciBench** features a structured and fine-grained taxonomy that categorizes materials science questions into 6 primary fields and 31 sub-fields, and includes a three-tier difficulty classification based on the reasoning length required to solve each question. **MatSciBench** provides detailed reference solutions enabling precise error analysis and incorporates multimodal reasoning through visual contexts in numerous questions. Evaluations of leading models reveal that even the highest-performing model, `Gemini-2.5-Pro`, achieves under 80% accuracy on college-level materials science questions, highlighting the complexity of **MatSciBench**. Our systematic analysis of different reasoning strategies—basic chain-of-thought, tool augmentation, and self-correction—demonstrates that no single method consistently excels across all scenarios. We further analyze performance by difficulty level, examine trade-offs between efficiency and accuracy, highlight the challenges inherent in multimodal reasoning tasks, analyze failure modes across LLMs and reasoning methods, and evaluate the influence of retrieval-augmented generation. **MatSciBench** thus establishes a comprehensive and solid benchmark for assessing and driving improvements in the scientific reasoning capabilities of LLMs within the materials science domain.

**Dataset:** huggingface.co/datasets/MatSciBench/MatSciBench
**Code:** anonymous.4open.science/r/MatSciBench

## 1 Introduction

Recent years have witnessed remarkable advancements of LLM reasoning abilities. From Chain of thought (Wei et al., 2022) to self-correction (Shinn et al., 2023) and tool-augmentation (Gou et al., 2023), the boundaries of LLM reasoning have expanded dramatically. What began with grade-school arithmetic calculations (Cobbe et al., 2021) has evolved to solving problems at the level of International Mathematical Olympiad (IMO) silver medalists (DeepMind, 2024). The o-series model of OpenAI's can even solve a substantial portion of frontier mathematical problems that would typically require hours of concentrated effort from expert mathematicians (OpenAI, 2025; Glazer et al., 2024).

Beyond LLMs' notable achievements in mathematics, general scientific reasoning has emerged as a new area of interest, where solving problems requires a proper combination of reasoning and domain-specific knowledge (Truhn et al., 2023; Ma et al., 2024a;b). Scientific reasoning benchmarks reveal that LLMs suffer from identifying correct scientific assumptions and often demonstrate flawed understanding of scientific formulas and principles (Wang et al., 2023). Those findings indicate that scientific reasoning presents unique challenges to LLMs compared to pure mathematical questions. Therefore, numerous benchmarks have been proposed towards assessing LLM's scientific reasoning capability, spanning from grade-school (Lu et al., 2022) to PhD-level (Feng et al., 2025) problems across domains (Huang et al., 2024a; Acharya et al., 2023).

Despite the abundance of scientific problem-solving benchmarks, LLMs' reasoning abilities in materials science remain underexplored. Materials science occupies a unique position at the intersection of physics and chemistry, bridging fundamental science and engineering applications. This interdisciplinary field inherently relies on knowledge integration across multiple domains and requires complex reasoning capabilities. Existing reasoning benchmarks in materials science are limited by the lack of comprehensive evaluation and correct solutions (Zaki et al., 2024), or by the dependence on synthetic data generated by LLMs themselves, which introduces unavoidable noises (Alampara et al., 2024). In addition, none of the existing benchmarks adequately assesses the multimodal reasoning ability of LLM in material science.

To comprehensively evaluate LLMs' reasoning abilities in materials science, we propose **MatSciBench**, a benchmark comprising 1340 meticulously curated questions from 10 college-level textbooks spanning essential subdisciplines of materials science. All questions are open-ended to prevent model guessing while enabling objective assessment through rule-based judgment. For structured evaluation, **MatSciBench** constructs a comprehensive and fine-grained taxonomy with 6 primary fields (Materials, Properties, Structures, Fundamental Mechanisms, Processes, Failure Mechanisms) and 31 sub-fields that capture materials science's interdisciplinary nature, enabling assessment of reasoning abilities on specific domains. In addition, questions are classified into three difficulty levels based on reasoning length required to solve the question, with 50.7% easy, 29.1% medium, and 20.1% hard questions. The 270 hard questions require long solving process, deliberately challenging models' complex reasoning capabilities. Detailed solutions to 944 of the questions are included to facilitate error categorization and process-level evaluations. The benchmark also incorporates 315 questions with visual contexts to assess multimodal reasoning abilities.

The o-series models from OpenAI, such as `o4-mini`, along with `Gemini-2.5-Pro`, `DeepSeek-R1`, `GPT-5`, `Claude-4-Sonnet`, and `Qwen3-235b-a22b-thinking`, represent a new class of LLMs that exhibit complex reasoning by generating extended intermediate outputs before producing final answers. These models are informally referred to as *thinking models* or reasoning models, distinguishing them from traditional LLMs like `GPT-4.1`, `Claude-3.7`, `DeepSeek-V3`, `Llama-4-Maverick`, and `Gemini-2.0-Flash`, classified as *non-thinking models* (Chen et al., 2025). We conduct extensive experiments on **MatSciBench** to evaluate and compare the reasoning capabilities of these six thinking models against five non-thinking models in materials science problem solving. In addition, we also evaluate the effectiveness of self-correction and tool-augmentation (i.e., integration of Python code) on non-thinking models in addition to the basic CoT. Our results indicate that while `Gemini-2.5-Pro` lead with approximately 77% accuracy, the best-performing non-thinking model, Llama-4-Maverick, achieves a comparable 71%. However, none of the techniques—basic CoT, self-correction, or tool-augmentation—consistently outperforms the others across all models, demonstrating that effectiveness depends significantly on the base model.

Our systematic analysis of LLM reasoning capabilities examine multiple dimensions: difficulty levels, reasoning efficiency, multimodal reasoning, and failure patterns. The key findings from our analysis include: (1) thinking models' performance is insensitive to question difficulty, suggesting that they better handle reasoning-intensive tasks; (2) performance improves with longer outputs, establishing a clear efficiency-accuracy trade-off frontier; (3) image-included questions lead to poorer performance in multimodal models compared to text-only questions on the same LLMs, highlighting the inherent challenges of multimodal reasoning; (4) by categorizing incorrectly answered responses into predefined error types,, we discovered that all tested models suffer from errors based on domain knowledge inaccuracies and question comprehension failures. Although the three reasoning methods are capable of reducing specific types of errors, they may concurrently amplify other types of errors; (5) our case study suggests that RAG may have limited effectiveness in reducing knowledge-based errors and could potentially contribute to increased hallucination rates.

Our contributions are listed as follows:

- We introduce **MatSciBench**, a comprehensive and challenging materials science reasoning benchmark comprising 1340 expert-curated questions from college-level textbooks across essential subdisciplines, featuring a structured taxonomy of 6 primary fields and 31 sub-fields, three-tier difficulty classification, detailed solutions for 944 questions, and 315 questions with visual contexts for multimodal reasoning evaluation.

- We benchmark SOTA LLMs, including six thinking models and five non-thinking models. Additionally, we enhance the non-thinking models with three popular reasoning methods. This provides the most comprehensive evaluation and comparison of reasoning capabilities in materials science across different models and methods.

- We present a comprehensive multi-dimensional analysis of LLM reasoning capability across difficulty levels, reasoning efficiency, accuracy trade-offs, multimodal reasoning capabilities, and failure patterns. We additionally conduct a case study exploring the influence of RAG on scientific reasoning in materials science. This thorough evaluation establishes a foundation for future improvements in scientific reasoning models.

## 2 RELATED WORK

### 2.1 BENCHMARKING LLM'S STEM PROBLEM SOLVING ABILITIES

As LLMs continue to develop reasoning abilities, solving scientific problems is considered a fundamental dimension and has been the focus of numerous benchmarks. GSM8K (Cobbe et al., 2021), MATH (Hendrycks et al., 2021), along with a series of benchmarks (Mirzadeh et al., 2024) evaluated the mathematical abilities of language models. With the emergence of multimodal LLMs, MathVista (Lu et al., 2023) further includes visual contexts to benchmark the multimodal reasoning abilities. With the growth of reasoning capabilities, competitive level questions like OlympiadBench (He et al., 2024) and PutnamBench (Tsoukalas et al., 2024), and advanced graduate-level math like Frontier Math (Glazer et al., 2024) and HARDMATH (Fan et al., 2024) set new standards for reasoning models.

Beyond mathematics, natural science questions involve not only reasoning but also domain knowledge, thus incentivizing increased interest, particularly in chemistry, physics, and biology (Welbl et al., 2017; Lu et al., 2022; Rein et al., 2024). SciBench (Wang et al., 2023), MMMU (Yue et al., 2024a), MMMU-Pro (Yue et al., 2024b) covers college-level scientific question solving requires both domain knowledge and sophisticated reasoning. OlympicArena (Huang et al., 2024b) contributes Olympiad-level, multimodal problems across seven scientific fields, and SuperGPQA (Du et al., 2025) further expands coverage to 285 graduate-level disciplines. Besides problem solving, SciEval (Sun et al., 2024), SciKnowEval (Feng et al., 2024) evaluate multi-level capabilities of LLM in scientific domain. In addition to those general natural scientific reasoning benchmarks, a series of works Acharya et al. (2023); Li et al. (2025a) focus on specific domains. PhysReason (Zhang et al., 2025), PHYSICS (Feng et al., 2025), MM-PhyQA (Anand et al., 2024) specialize on the physical questions; ChemEval (Huang et al., 2024a) benchmarks chemistry abilities; Sarwal et al. for Bioinformatics; Meshram et al. (2024) for electronics.

### 2.2 AI FOR MATERIAL SCIENCE

**Materials Databases.** Well-curated data repositories form the foundation of modern materials informatics. The Materials Project (Jain et al., 2013) pioneered this approach with its extensive catalog of computed properties, establishing a framework now expanded by complementary initiatives like NOMAD (Draxl and Scheffler, 2019) and AFLOW (Curtarolo et al., 2012). These platforms leverage FAIR principles (Wilkinson et al., 2016) to ensure data quality and accessibility—essential prerequisites for meaningful AI applications. The breadth and depth of these resources have dramatically reduced the barriers to computational materials exploration.

**LLMs in Materials Science.** Large Language Models (LLMs) are rapidly emerging as versatile and powerful tools within the materials science domain. Zhang et al. (2024) demonstrated how LLMs can function as coordinating agents, breaking down complex materials challenges and orchestrating specialized computational tools. Beyond this organizational role, LLMs excel at extracting insights from scientific literature and suggesting novel experimental approaches (Jablonka et al., 2023). Perhaps most intriguingly, Gruver et al. (2024) showed that fine-tuned language models can generate valid crystal structures directly as text. To systematically advance these capabilities, benchmarks like LLM4Mat-Bench (Rubungo et al., 2024) provide crucial evaluation frameworks that help refine these models for materials-specific tasks.

## 3 DATASET

### 3.1 DATA COLLECTION AND PROCESSING

For our materials science benchmark dataset, we curated a collection of problems from textbooks across multiple sub-fields. We selected widely-adopted undergraduate and graduate textbooks that include both comprehensive references (like "Fundamentals of Materials") and specialized resources focusing on specific domains (such as "Electronic Magnetic and Optical Materials"). The choice of textbooks was guided and validated by materials science experts. We first identified the major subfields of materials science and then selected textbooks in these areas that provide exercise solutions and are accessible online. These sources collectively provide diverse problem types that cover the breadth of materials science concepts. A full set of textbooks details are provided in Appendix B.

We used Mistral optical character recognition (OCR) (Mistral AI Team, 2025) to digitize both textual and visual content of these textbooks. Then we implemented a parsing algorithm to identify the example problems and solutions from the digital copies. Each question-answer pair was structured into a standardized format. Following the initial extraction, each entry was manually reviewed and corrected by domain experts to ensure accuracy and completeness. We applied strict filtering criteria, retaining only questions with determinate answers in the form of numerical values or formulas.

### 3.2 DATASET STATISTICS

Our benchmark comprises 1340 question-answer pairs structured in a standardized format. Each entry contains fields for question text, solution text, associated images, difficulty level, domain classification, and problem type. Questions are categorized as either numerical or formula type according to the answer, with 92.4% being numerical and 7.6% requiring formula derivation, 315 questions (23.5%) include images.

### 3.3 TAXONOMY CLASSIFICATION

We developed a comprehensive hierarchical taxonomy to systematically categorize questions across fundamental materials science domains. Our taxonomy design was informed by established materials science curricula and reference texts, including Shackelford (2015); Shackelford et al. (2016); Ashby et al. (2019). The taxonomy framework reflects both the traditional organization of materials science education and the critical concepts that underpin fundamental mechanisms at different length scales. The taxonomy consists of six primary fields, each containing detailed subcategories:

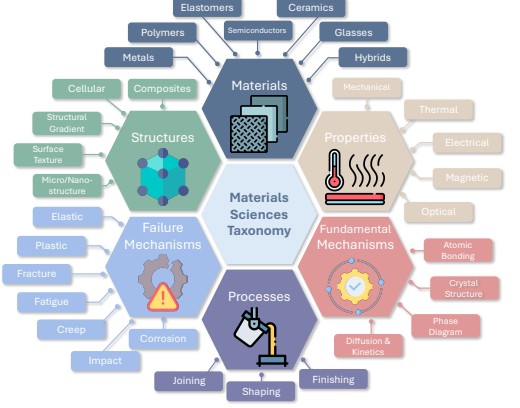

Figure 1: Taxonomy of **MatSciBench** Materials Science QAs.

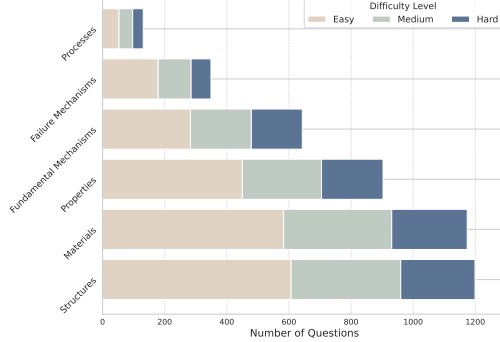

Figure 2: Difficulty Distribution by Taxonomy Primary Fields.

1. **Materials**: Metals, Polymers, Elastomers, Semiconductors, Ceramics, Glasses, Hybrids
2. **Properties**: Mechanical, Thermal, Electrical, Magnetic, Optical
3. **Structures**: Composites, Cellular, Structural Gradient, Surface Texture, Micro/Nano-structure

4. **Fundamental Mechanisms**: Atomic Bonding, Crystal Structure, Phase Diagram, Diffusion & Kinetics
5. **Processes**: Joining, Shaping, Finishing
6. **Failure Mechanisms**: Elastic, Plastic, Fracture, Fatigue, Creep, Impact, Corrosion

Figure 1 shows our manually developed taxonomy that covers the domains of QAs collected in **MatSciBench**. The taxonomy consists of six primary fields: Materials, Properties, Structures, Fundamental Mechanisms, Processes, and Failure Mechanisms, each containing detailed subcategories. This multi-dimensional classification scheme enables us to capture the interdisciplinary nature of materials science problems, where a single question might span multiple domains. Our taxonomy not only provides a nuanced understanding of the dataset composition but also enables targeted evaluation of model performance across specific subfields and their intersections, offering insights into how AI models handle different aspects of materials science knowledge. Additional details of dataset, including the data leakage detection, can be found in Appendix B.

### 3.4 DIFFICULTY CLASSIFICATION

We implemented a three-tier classification from easy, medium to hard, to assess question difficulty. Difficulty is assessed with response lengths from `Claude-3.7-Sonnet`, which classifies questions into Easy (50.7%), Medium (29.1%), and Hard (20.1%) categories based on the length of model responses required to solve them. This distribution provides a balanced representation across difficulty levels, while differentiating questions in terms of the knowledge and reasoning length required to derive a correct solution. We validated this length-based approach across multiple models and consistently observed that accuracy decreases while response length increases with difficulty, confirming the reliability of this assessment method. To verify the robustness of our classification, we use step-count analysis based on the judgment of `Gemini-2.0-Flash` for solution steps required to solve each question, along with additional pattern-based and KNN-based validation methods. Details of the validation approaches are discussed in Appendix B.

## 4 EXPERIMENTS

### 4.1 MODELS AND METHODS

For proprietary models, we evaluate `GPT-4.1` (OpenAI, 2025), `Claude-3.7-Sonnet` (Anthropic, 2025), `Gemini-2.0-Flash` (Google DeepMind, 2024), and the thinking models `o4-mini` (OpenAI, 2024), `Gemini-2.5-Pro` (Google DeepMind, 2025), `GPT-5` (OpenAI, 2025), `Claude-Sonnet-4` (Anthropic, 2025); for open-weight models, we evaluate `DeepSeek-V3` (Liu et al., 2024), `llama-4-maverick` (Meta AI, 2025), and the thinking models `DeepSeek-R1` (Guo et al., 2025), `Qwen3-235b-a22b-thinking` (Yang et al., 2025). Among these models, `GPT-4.1`, `Claude-3.7-Sonnet`, `Gemini-2.0-Flash`, `o4-mini`, `Gemini-2.5-Pro`, `GPT-5`, and `Claude-Sonnet-4` support visual inputs.

For non-thinking models, we adapt three prompting methods: basic CoT, self-correction, and tool-augmentation. The self-correction methods follows Huang et al. (2023); Kim et al. (2023); Shinn et al. (2023), invoking 3 rounds of conversation with the model: (1) the initial response, (2) detecting issues in the initial attempt, and (3) revising the initial attempt based on the detected problem. The tool-augmentation method prompts the model to generate Python code, executes it using a code interpreter (Gou et al., 2023; Yang et al., 2024a), and derives the final answer based on the execution results. The detailed prompts are provided in the Appendix C.

### 4.2 EVALUATION

The correctness of the output answers is evaluated using a hybrid approach that combines rule-based evaluation and LLM-based evaluation. We adapt the rule-based evaluation system from Qwen-2.5 Math (Yang et al., 2024a). Following the previous works (Methani et al., 2020; Gupta et al., 2024), we apply a relaxed numerical tolerance of 5% to account for approximation errors in calculations and image recognition. To address the limitations of rule-based systems in handling complex formulas and equations, we supplement this approach with `Gemini-2.0-Flash` for formula-type questions.

Table 1: Experimental Results in Terms of Accuracy Score (%) on **MatSciBench**(questions w/o images). **Bold** indicates the best performance, and Underline indicates the second best.

| Model | Failure | Fund. | Materials | Proc. | Prop. | Struct. | Overall |
|---|---|---|---|---|---|---|---|
| *Non-Thinking Models* | | | | | | | |
| **Claude-3.7-Sonnet** | 65.66 | 65.97 | 65.89 | 63.64 | 64.84 | 68.74 | 67.32 |
| *+Correction* | 66.79 | 63.87 | 67.11 | 64.65 | 65.42 | 69.51 | 68.00 |
| *+Tool* | **72.08** | 66.18 | 70.75 | 64.65 | **70.89** | 72.35 | 71.51 |
| **DeepSeek-V3** | 62.64 | 61.97 | 65.67 | 63.64 | 63.26 | 66.89 | 66.15 |
| *+Correction* | 67.17 | 60.50 | 62.91 | 63.64 | 63.26 | 65.46 | 64.39 |
| *+Tool* | 61.51 | 59.03 | 62.69 | 62.63 | 57.93 | 64.15 | 62.44 |
| **Gemini-2.0-Flash** | 60.75 | 55.04 | 59.71 | 52.53 | 58.36 | 60.00 | 59.90 |
| *+Correction* | 59.62 | 59.24 | 61.92 | 51.52 | 59.22 | 63.28 | 62.34 |
| *+Tool* | 67.55 | 65.97 | 68.65 | 69.70 | 68.30 | 70.49 | 69.46 |
| **GPT-4.1** | 65.66 | **68.91** | 70.42 | 61.62 | 67.58 | 71.80 | 70.73 |
| *+Correction* | 66.04 | 65.13 | 68.10 | 57.58 | 65.56 | 69.29 | 68.00 |
| *+Tool* | 63.02 | 62.18 | 61.92 | 55.56 | 60.81 | 62.62 | 61.66 |
| **Llama-4-Maverick** | 69.06 | **68.91** | **71.30** | **72.73** | 69.16 | **73.11** | **71.61** |
| *+Correction* | 69.43 | 66.18 | 69.87 | 70.71 | 68.16 | 71.15 | 69.95 |
| *+Tool* | 68.30 | 63.24 | 68.21 | 67.68 | 65.27 | 68.85 | 68.20 |
| *Thinking Models* | | | | | | | |
| **Claude-4-Sonnet** | 58.49 | 52.52 | 54.86 | 56.57 | 52.31 | 54.64 | 54.44 |
| **DeepSeek-R1** | 71.70 | 71.43 | 73.84 | 74.75 | 72.62 | 75.30 | 73.95 |
| **Gemini-2.5-Pro** | **78.49** | **76.89** | **77.15** | **75.76** | **74.50** | **78.69** | **77.37** |
| **Qwen3-235B** | 73.58 | 69.96 | 71.96 | 68.69 | 70.17 | 73.33 | 72.10 |
| **GPT-5** | 67.17 | 64.71 | 65.34 | 67.68 | 62.97 | 65.57 | 64.88 |
| **o4-mini** | 72.08 | 73.32 | 73.73 | 69.70 | 72.91 | 74.97 | 74.34 |

Table 2: Experimental Results in Terms of Accuracy Score (%) on **MatSciBench** (questions w/ images). **Bold** indicates the best performance, and Underline indicates the second best.

| Model | Failure | Fund. | Materials | Proc. | Prop. | Struct. | Overall |
|---|---|---|---|---|---|---|---|
| Claude-3.7-Sonnet | 29.76 | 37.13 | 37.31 | 40.62 | 34.45 | 34.51 | 34.60 |
| Claude-Sonnet-4 | 30.95 | 40.12 | 39.18 | 46.88 | 35.89 | 38.73 | 37.46 |
| Gemini-2.0-Flash | 25.00 | 32.34 | 26.49 | 28.12 | 27.27 | 26.41 | 26.03 |
| Gemini-2.5-Pro | 39.29 | 40.12 | 42.16 | 31.25 | 41.63 | 38.73 | 39.05 |
| GPT-5 | **42.86** | **53.89** | **49.63** | **59.38** | **46.89** | **50.00** | **48.89** |
| o4-mini | 33.33 | 40.72 | 37.69 | 43.75 | 36.36 | 37.32 | 37.14 |

The LLM's judgment serves as the final determinant of correctness for these complex mathematical expressions. The performance in terms of accuracy score of all models on text-only questions is presented in Table 1, and the performance of multimodal models on images-included questions is presented in Table 2. We also evaluate a range of small and medium scale models and repeat the runs to confirm that the results are largely deterministic, as reported in Appendix D.

## 4.3 RESULTS

**Observation 1.** Among non-thinking models, `Llama-4-Maverick` achieves the best overall accuracy (71.61%) under the basic chain-of-thought (CoT) setting. `GPT-4.1` ranks second (70.73%) in the basic CoT category, although its performance decreases when tools are introduced. `Claude-3.7-Sonnet` shows relatively lower accuracy with basic CoT (67.32%), but improves to 71.51% with tool integration, becoming the second-best performer in the tool-augmented setting. `Gemini-2.0-Flash` has bad accuracy under the basic CoT condition (59.90%) but substantially improves with tool use, reaching 69.46%. For thinking models, `Gemini-2.5-Pro` attains the best results overall with 77.37%, surpassing all other models. `DeepSeek-R1` is the strongest among open-weight thinking models with 73.95%, closely followed by `Qwen3-235B` (72.10%). These results indicate that the performance gap between open-weight and proprietary models is narrowing.

**Observation 2.** No single prompting method demonstrates consistently superior performance across all models. The performance improvements achieved through tool-augmentation varies significantly between models: `Claude-3.7` and `Gemini-2.0-Flash` show substantial increases in overall performance, `GPT-4.1`, `DeepSeek-V3`, and `Llama-4-Maverick` exhibits performance degradation. The self-correction technique generally decreases performance across most models, converting more correct answers to incorrect ones than vice versa. Only `Gemini-2.0-Flash` shows substantial performance improvements under this approach.

**Observation 3.** In the multimodal evaluation, `GPT-5` delivers the strongest performance, achieving the highest overall accuracy (48.89%) and leading across all individual categories. `Gemini-2.5-Pro` ranks second overall with 39.05%. Claude models exhibit moderate performance, with `Claude-Sonnet-4` (37.46%) slightly surpassing `Claude-3.7-Sonnet` (34.60%). The `o4-mini` model achieves a comparable score (37.14%) to `Claude-Sonnet-4`, but remains behind `GPT-5` and `Gemini-2.5-Pro`. These results highlight the superiority of `GPT-5` in handling multimodal reasoning tasks.

## 5 ANALYSIS

### 5.1 PERFORMANCE ACROSS DIFFICULTY LEVELS

The accuracy scores of different models across difficulty levels are shown in Figure 3. Most models exhibit expected performance degradation patterns with increasing difficulty, suggesting that complex reasoning process prevent them from reaching correct answers. `o4-mini` shows an interesting pattern: its accuracy on hard questions is not lower than on medium questions. This pattern may suggest that, for this small-scale reasoning-focused model, the main difficulty might not lie in the length of reasoning, but rather in domain knowledge.

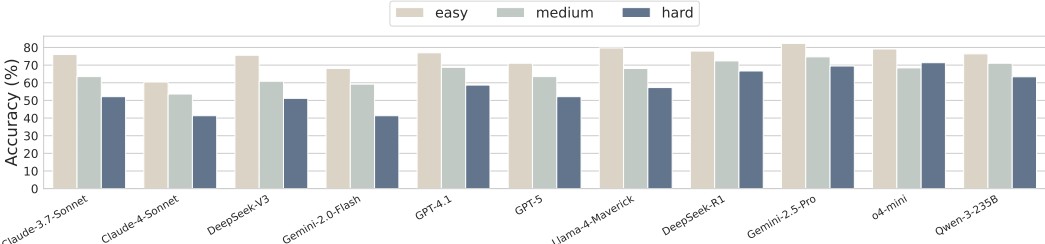

Figure 3: The Performance of LLMs across Difficulty Levels.

## 5.2 EFFICIENCY V.S. ACCURACY

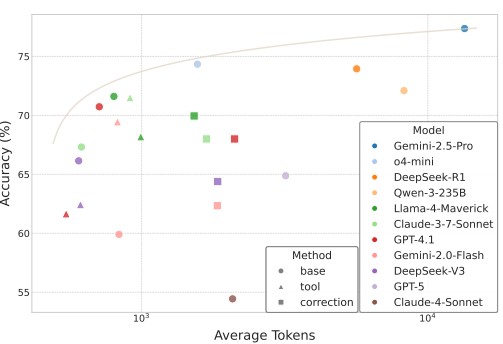 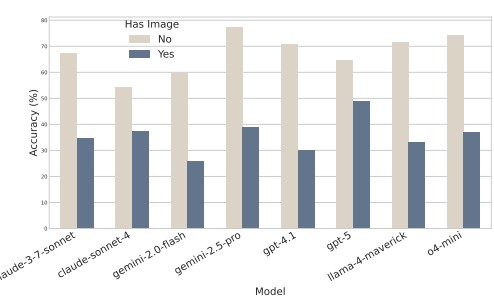

Figure 4: The Average Output Length v.s. The Accuracies.

Figure 5: The Performance Comparison of MLLM between Questions w and w/o Images.

Thinking models often generate highly verbose outputs. This verbosity frequently involves branching, backtracking, error validation, and correction (Yeo et al., 2025), which, although beneficial for arriving at correct results, may compromise efficiency. This underscores a fundamental trade-off between reasoning accuracy and efficiency.

Figure 4 illustrates the relationship between performance and output length by showing token usage across different models and methods, with the boundary line representing the reasoning efficiency frontier. When using basic CoT prompting, thinking models consume significantly more tokens while achieving superior performance compared to non-thinking models. Self-correction prompting substantially increases output length without consistently improving performance—sometimes even degrading results. In contrast, tool augmentation provides a more economical approach, requiring minimal additional tokens while boosting performance across many models.

## 5.3 PERFORMANCE DROP DUE TO VISUAL CONTEXT

Image-included questions are significantly more challenging than text-only questions for multimodal LLMs, with a significantly lower accuracy scores, as presented in Figure 5. We identified two major sources of error in questions involving visual context: (i) many images in materials science are inherently three-dimensional—such as lattice cells or atomic arrangements—which challenge the spatial reasoning abilities of multimodal LLMs; (ii) many figures are diagrams or plots that require models to extract numerical values precisely, a task that remains difficult for current multimodal LLMs.

## 5.4 FAILURE PATTERN

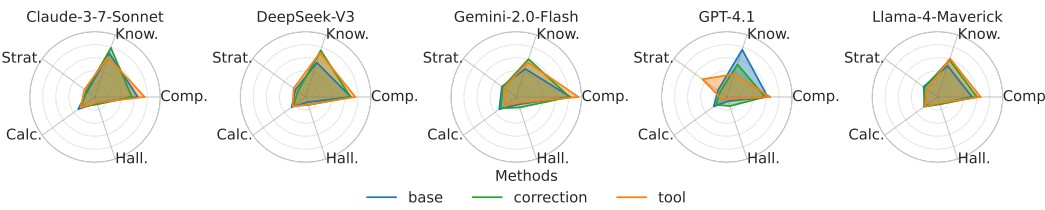

Figure 6: Error categorization for non-thinking models. Types of errors are problem comprehension deficiencies (Comp.), domain knowledge gaps (Know.), flawed solution strategies (Strat.), calculation inaccuracies (Calc.), and hallucinated content (Hall.).

To investigate the challenges LLMs face in solving materials science problems, we manually examine incorrect responses and identified five major error categories: problem comprehension deficiencies, domain knowledge gaps, flawed solution strategies, calculation inaccuracies, and hallucinated content. To conduct systematic analysis of these error patterns automatically, we em-

ployed `Gemini-2.0-Flash` to categorize mistakes across these five categories, evaluating all non-thinking models and prompting methods on text-only questions with reference solutions. When multiple errors exist, we classify them into the first appearing one in the solution. Detailed prompts and definitions for each category are provided in Appendix E.

The error rates across categories are presented in Figure 6. These findings reveal consistent patterns across all models, with deficiencies in domain knowledge and question comprehension representing the most critical limitations—exceeding even calculation errors. While errors caused by hallucinations are still present, they occur less frequently than other error types. As expected, tool-augmentation methods reduced numerical errors across all models, with the most significant improvements observed in `Gemini-2.0-Flash`. Self-correction methods, on the other hand, did not provide consistent improvements across any of the tested models in any error category.

## 5.5 RETRIEVAL AUGMENTED GENERATION: A CASE STUDY

Retrieval Augmented Generation (RAG) has long been regarded as an effective approach to enhance model performance in scientific domains where specialized knowledge is necessary for completing tasks (Lála et al., 2023; Li et al., 2025b). To verify this approach on material science reasoning tasks, we conducted a case study using `DeepSeek-V3` on **MatSciBench**. We implemented RAG through web searching: given a question, the LLM formulates a search query, retrieves up to five most relevant results from the Tavily API, summarizes the most useful information, and appends this to the original question. The failure pattern when using RAG is presented in Figure 7. Surprisingly, RAG does

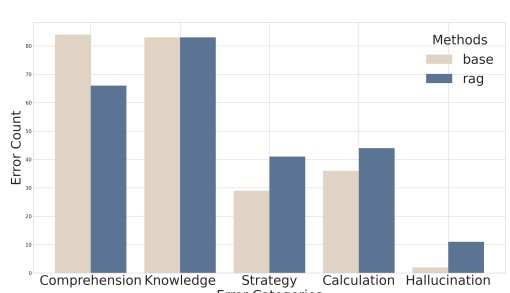

Figure 7: The Error Categories Counts of Basic CoT and RAG.

not reduce knowledge-related errors but instead improves problem comprehension. We hypothesize that web searching doesn't consistently retrieve correct and useful information, thus fail to reliably enhance knowledge accuracy and occasionally even inducing hallucination. However, the additional contextual information may help the model better comprehend questions and identify relevant information for solving them. Specific examples for both cases, and an additional RAG case study for Gemini-2.0-Flash, can be found in Appendix E.

## 6 CONCLUSION AND LIMITATION

In this work, we present **MatSciBench**, a benchmark comprising 1340 college-level materials science questions spanning all essential subdisciplines. We evaluate state-of-the-art thinking and non-thinking models on **MatSciBench**, employing three different reasoning methods for non-thinking models. Our results reveal significant performance discrepancies among LLMs on materials science reasoning tasks and highlight the varying effectiveness of different reasoning approaches. We further analyze model performance across multiple dimensions: difficulty levels, reasoning efficiency, multimodal reasoning capabilities, failure patterns, and retrieval-augmented generation (RAG). This comprehensive analysis enhances our understanding of model performance and establishes a foundation for further improvements in materials science reasoning capabilities.

While comprehensive, **MatSciBench** has several limitations. First, the benchmark would benefit from a larger dataset size; however, the intensive human labor required to create high-quality, real-data benchmarks makes scaling difficult. In addition, due to financial constraints, we were unable to benchmark some most advanced models(e.g., `o3`, `Claude-Opus-4.1`).

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

## USAGE OF LARGE LANGUAGE MODELS

In this work, besides running LLMs in experiments, we use LLMs for the following purposes:

1. Aid or Polish Writing (Gemini, ChatGPT, Claude)

2. Literature Retrieval and Discovery (e.g., finding related work) (Gemini Deep Research, ChatGPT Deep Research)

3. Assisting Code Writing and Debugging (Claude Code, Cursor(GPT, Claude))

We fully understand the responsibility of using LLMs in academic research. We carefully monitor any potential problems, such as plagiarism or scientific misconduct (e.g., fabrication of facts) when using LLMs. We make sure these problems do not occur in the paper.

## CONTENTS

## A  ADDITIONAL RELATED WORK

### A.1  AI FOR MATERIAL SCIENCE

**AI-Powered Material Design and Simulation.** The materials science toolkit continues to expand with specialized AI approaches that complement language models. Generative frameworks developed by Zeni et al. (2025), Xie et al. (2022), and Yang et al. (2024b) create entirely new materials with tailored properties, while machine learning potentials from Chen and Ong (2022), Deng et al. (2023), and Merchant et al. (2023) deliver quantum-accurate simulations at a fraction of traditional computational costs. These advances enable rapid screening of materials candidates that would be impractical using conventional methods.

The integration of AI continues to push scientific boundaries, notably with the rise of self-driving laboratories (MacLeod et al., 2020; Reker et al., 2023) that automate and accelerate the experimental discovery cycle. Concurrently, the need to understand and trust these sophisticated models has spurred the development of explainable AI techniques tailored for materials science (Rickman et al., 2019; Liao et al., 2022). The synergy between comprehensive data resources, the multifaceted capabilities of LLMs, advanced generative and predictive algorithms, and emerging autonomous and interpretable systems heralds a new, accelerated era of materials innovation with profound implications for technology and society.

## B  ADDITIONAL DETAILS ABOUT DATASET

### B.1  QUESTION SOURCE

We list the source of our questions in Table 3.

Table 3: Source Textbooks Used for Question-answer Collection

| Textbook | Author(s) | # QAs |
|---|---|---|
| Introduction to Materials Science for Engineers | James F. Shackelford | 349 |
| The Science and Engineering of Materials | Donald R. Askeland, Pradeep P. Fulay, and Wendelin J. Wright | 287 |
| Materials Science and Engineering: An Introduction | William D. Callister, Jr. | 61 |
| Fundamentals of Materials Science and Engineering: An Integrated Approach | William D. Callister, Jr. and David G. Rethwisch | 393 |
| Mechanical Behavior of Materials | William F. Hosford | 83 |
| Electronic, Magnetic, and Optical Materials | Pradeep P. Fulay and Jung-Kun Lee | 72 |
| Materials and Process Selection for Engineering Design | Mahmoud M. Farag | 27 |
| Fundamentals of Ceramics | Michel W. Barsoum | 29 |
| Physical Metallurgy | William F. Hosford | 27 |
| Polymer Science and Technology | Joel R. Fried | 12 |
| **Total** | | **1340** |

## B.2 DATA EDITING UI

The user interface of our data editing app is presented in Figure 8. This UI present the QA and allow users to edit each field of the QA.

## B.3 EXAMPLE OF QUESTION

Here we present an example from our dataset with all its attributes.

---

**Introduction to Materials Science for Engineers Example 9.12**

**Question**

A fireclay refractory ceramic can be made by heating the raw material kaolinite, $Al_2(Si_2O_5)(OH)_4$, thus driving off the waters of hydration. Determine the phases present, their compositions, and their amounts for the resulting microstructure (below the eutectic temperature). Give your answer as a tuple (Amount of $SiO_2$ in mol%, Amount of Mullite in mol%).

**Solution**

A modest rearrangement of the kaolinite formula helps to clarify the production of this ceramic product:

$$Al_2(Si_2O_5)(OH)_4 = Al_2O_3 \cdot 2SiO_2 \cdot 2H_2O$$

The firing operation yields

$$Al_2O_3 \cdot 2SiO_2 \cdot 2H_2O \xrightarrow{\text{heat}} Al_2O_3 \cdot 2SiO_2 + 2H_2O \uparrow$$

The remaining solid, then, has an overall composition of

$$mol\%Al_2O_3 = \frac{mol\,Al_2O_3}{mol\,Al_2O_3 + mol\,SiO_2} \times 100\%$$

$$= \frac{1}{1+2} \times 100\% = 33.3\%$$

Using Figure 9.23, we see that the overall composition falls in the $SiO_2+$ mullite two-phase region below the eutectic temperature. The $SiO_2$ composition is 0 mol%$Al_2O_3$ (i.e., 100%$SiO_2$). The composition of mullite is 60 mol%$Al_2O_3$.

---

**Fundamentals_of_Materials_Instructors_Example 6.37**

**Question Source**

A diffusion couple similar to that shown in figure 6.1a is prepared using two hypothetical metals $A$ and $B$. After a 20-h heat treatment at $800^\circ$C (and subsequently cooling to room temperature) the concentration of $B$ in $A$ is 2.5 wt% at the 5.0 mm position within metal $A$. If another heat treatment is conducted on an identical diffusion couple, but at $1000^\circ$C for 20 h , at what position will the composition be 2.5 wt% $B$ ? Assume that the preexponential and activation energy for the diffusion coefficient are $1.5 \times 10^{-4}$ m$^2$/s and $125,000$ J/mol, respectively.

**Question Preview**

A diffusion couple similar to that shown in figure 6.1a is prepared using two hypothetical metals $A$ and $B$. After a 20-h heat treatment at $800^\circ$C (and subsequently cooling to room temperature) the concentration of $B$ in $A$ is 2.5 wt% at the 5.0 mm position within metal $A$. If another heat treatment is conducted on an identical diffusion couple, but at $1000^\circ$C for 20 h , at what position will the composition be 2.5 wt% $B$ ? Assume that the preexponential and activation energy for the diffusion coefficient are $1.5 \times 10^{-4}$ m$^2$/s and $125,000$ J/mol, respectively.

**Solution Source**

In order to determine the position within the diffusion couple at which the concentration of A in B is 2.5 B\mathrm{wt} (%)\$, we must employ equation \frac{x^2}{D t}=\mathrm{constant}\$ with B\$ constant. That is

$$ \frac{x^2}{D}=\text { constant } $$

Or

$$ \frac{x(800)^2}{D(800)}=\frac{x(1000)^2}{D(1000)} $$

It is first necessary to compute values for both $D(800)$ and $D(1000)$; this is accomplished using equation (D=D_0 \exp \left(-\frac{Q_d}{R T}\right)) as follows:

**Solution Preview**

In order to determine the position within the diffusion couple at which the concentration of A in B is 2.5 wt%, we must employ equation $\frac{x^2}{D t}=\mathrm{constant}$ with $t$ constant. That is

$$ \frac{x^2}{D} = \text{constant} $$

Or

$$ \frac{x800^2}{D800} = \frac{x1000^2}{D1000} $$

It is first necessary to compute values for both $D800$ and $D1000$; this is accomplished using equation $(D=D\_0 \exp \left(-\frac{Q\_d}{R T}\right))$ as follows:

$$ D800 = (1.5 \times 10^{-4}\ \text{m}^2/\text{s}) \exp\left[-\frac{125,000\ \text{J/mol}}{(8.31\ \text{J/mol}-\text{K})(800+273\ \text{K})}\right] $$
$$ 1.22 \times 10^{-10}\ \text{m}^2/\text{s} $$

$$ D1000 = (1.5 \times 10^{-4}\ \text{m}^2/\text{s}) \exp\left[-\frac{125,000\ \text{J/mol}}{(8.31\ \text{J/mol}-\text{K})(1000+273\ \text{K})}\right] $$
$$ 1.11 \times 10^{-9}\ \text{m}^2/\text{s} $$

Now, solving equation $(\frac{x\_{800}^2}{D\_{800}}=\frac{x\_{1000}^2}{D\_{1000}})$ (above) for $x1000$ yields

$$ x1000 = x800\sqrt{\frac{D1000}{D800}} $$
$$ = (5\ \text{mm})\sqrt{\frac{1.11 \times 10^{-9}\ \text{m}^2/\text{s}}{1.22 \times 10^{-10}\ \text{m}^2/\text{s}}} $$
$$ = 15.1\ \text{mm} $$

**Answer Source**

15.1

**Answer Preview**

15.1

**Unit**

mm

**Unit Preview**

mm

**Categories**

| Materials | Properties | Structures |
|---|---|---|
| Metals | Thermal | Composites |
| Fundamental Mechanisms | Processes | Failure Mechanisms |
| | | |

**Type**

N/A

**Number of Answers**

Single

Save & Recompile

Previous   Next   Delete Question

**Images**

**Figure 6.1** (a) A copper–nickel diffusion couple before a high-temperature heat treatment. (b) Schematic representations of Cu (red circles) and Ni (blue circles) atom locations within the diffusion couple. (c) Concentrations of copper and nickel as a function of position across the couple.

Image path: Images/Fundamentals_of_Materials_Instructors_Example 6.37_1.jpg
Remove Image

**Upload New Image**

Choose File   no file selected

**Notes**

**Notes Preview**

Figure 8: UI of Data Editing App

Using Equations $\frac{m_\alpha}{m_\alpha + m_\beta} = \frac{x_\beta - x}{x_\beta - x_\alpha}$ and $(\frac{m_\beta}{m_\alpha + m_\beta} = \frac{x - x_\alpha}{x_\beta - x_\alpha})$ yields

$$ \text{mol}\%\text{SiO}_2 = \frac{x_\text{mullite} - x}{x_\text{mullite} - x_\text{SiO}_2} \times 100\% = \frac{60 - 33.3}{60 - 0} \times 100\% $$
$$ = 44.5\ \text{mol}\% $$

and

$$\text{mol\,\% mullite} = \frac{x - x_{\text{SiO}_2}}{x_{\text{mullite}} - x_{\text{SiO}_2}} \times 100\% = \frac{33.3 - 0}{60 - 0} \times 100\%$$
$$= 55.5 \text{ mol\%}$$

**Image**

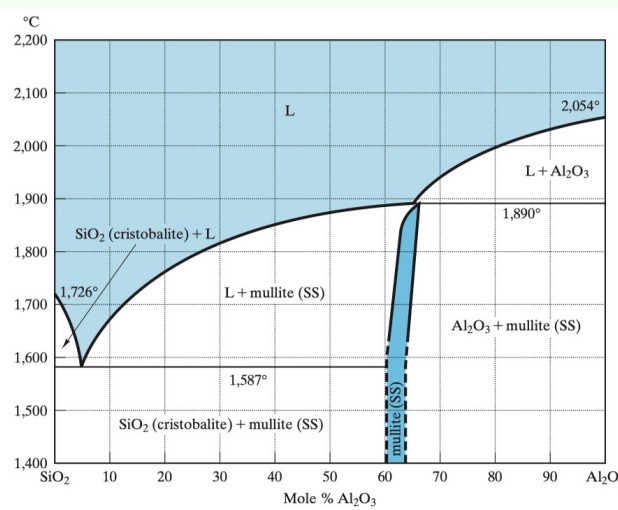

**FIGURE 9.23** $Al_2O_3$–$SiO_2$ *phase diagram. Mullite is an intermediate compound with ideal stoichiometry* $3Al_2O_3 \cdot 2SiO_2$. *(After F. J. Klug, S. Prochazka, and R. H. Doremus, J. Am. Ceram. Soc. 70, 750, 1987.) (Reprinted with permission of The American Ceramic Society (www.ceramics.org). All rights reserved.)*

**Answer**   (44.5, 55.5)
**Unit**   (mol%, mol%)
**Categories**
Materials: Ceramics
Properties: Thermal
Structures: Micro/Nano-structure
**Type**   NUM
**Number of Answers**   Multiple
**Difficulty Level**   medium

## B.4   TAXONOMY TREE

The proposed taxonomy for materials science QAs includes 6 major fields. For each question, we assign a six-digit "category vector" where each position corresponds to one of these primary fields. The digit in each position indicates the relevant subcategory within that field, with 0 representing "None" (indicating the field is not relevant to the question). For example, a question with category vector "310001" would be about Metals, Mechanical properties, with no relevant Structures, Fundamental Mechanisms, or Processes components, and related to Elastic deformation. Only 10 questions are not matched with any subfiled and are manually assigned to "Mechanical Properties" ("010000").

Figure 9 shows the distribution across taxonomy subcategories. This multi-dimensional classification scheme enables us to capture the interdisciplinary nature of materials science problems, where a single question might span multiple domains. Only 10 questions are not matched with any subfiled and are manually assigned to "Mechanical Properties" ("010000").

## B.5   COMPARISON OF DIFFICULTY ASSESSMENT METHODS

### B.5.1   RESPONSE-LENGTH BASED DIFFICULTY CLASSIFICATION

We evaluate question difficulty based on response lengths from a strong baseline model, `Claude-3.7-Sonnet`. The questions are classified into three tiers: Easy (50.2%), Medium

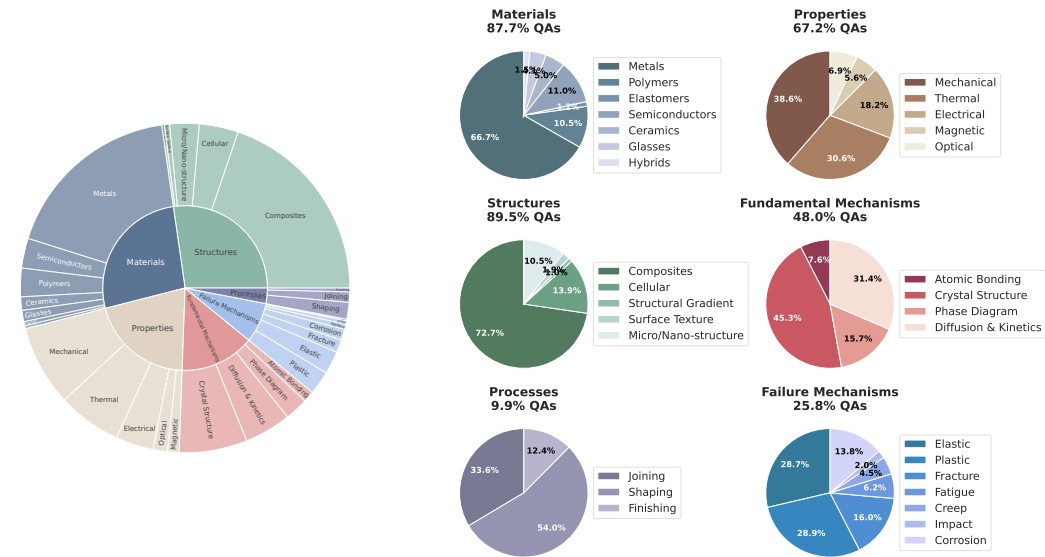

Figure 9: Taxonomy and QA distribution across each fields and sub-fields.

(29.3%) and Hard (20.5%). We applied this classification to other models to demonstrate its robustness. As shown in Table 4, for nearly every model, accuracy decreases while response length (characters) increases with difficulty, confirming the reliability of this assessment method.

Table 4: Model Performance and Response Length across Difficulty Levels Based on Response-Length Classification

| Model | Accuracy | | | Response Length | | |
|---|---|---|---|---|---|---|
| | Easy | Medium | Hard | Easy | Medium | Hard |
| claude-3-7-sonnet-20250219 | 0.640 | 0.541 | 0.480 | 1180 | 1496 | 1947 |
| deepseek-V3 | 0.712 | 0.558 | 0.487 | 1413 | 1690 | 2310 |
| deepseek-reasoner | 0.749 | 0.658 | 0.592 | 15901 | 18417 | 20689 |
| gemini-2.0-flash | 0.552 | 0.484 | 0.363 | 1713 | 2316 | 2816 |
| gemini-2.5-pro-preview-05-06 | 0.632 | 0.551 | 0.527 | 3209 | 3593 | 4331 |
| gpt-4.1-2025-04-14 | 0.591 | 0.506 | 0.414 | 1382 | 1786 | 2300 |
| llama-4-maverick | 0.693 | 0.553 | 0.495 | 2157 | 2626 | 3096 |
| o4-mini-2025-04-16 | 0.586 | 0.471 | 0.498 | 400 | 467 | 505 |
| qwen3-235b-a22b | 0.745 | 0.657 | 0.563 | 1640 | 1917 | 2188 |

The consistent pattern across multiple models validates this classification methodology, showing that longer required responses correspond to more complex reasoning requirements. We adopt this response-length based classification as our primary difficulty assessment method, with step-count analysis serving as validation.

### B.5.2   STEP-COUNT BASED DIFFICULTY CLASSIFICATION

To ensure robustness in our difficulty classification, we employed three complementary approaches to determine solution step counts:

1. **Pattern-based method**: This approach identifies explicit step indicators in solution text, such as numbered steps, paragraph breaks, and calculation indicators.

2. **KNN-based method**: Trained on 100 sampled examples, this approach uses few-shot learning to identify implicit solution steps even when they aren't explicitly numbered.

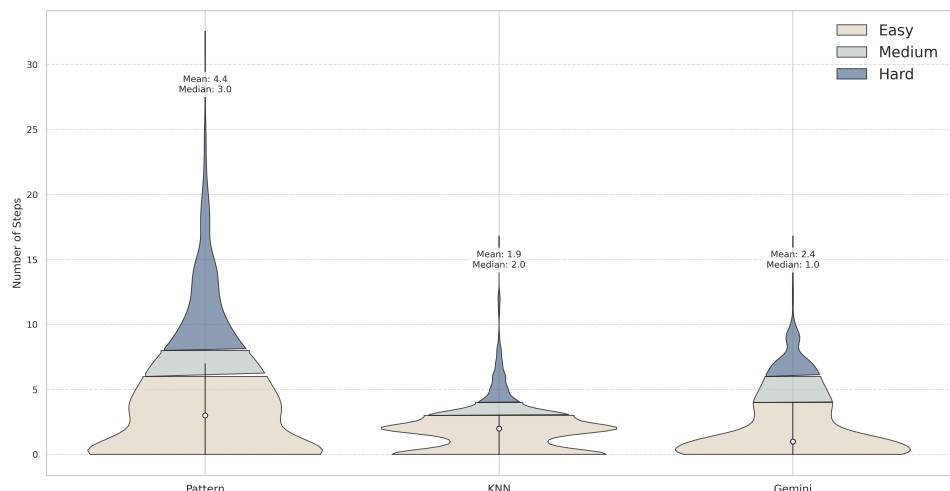

Figure 10: Distribution of solution steps count across the dataset.

3. **Gemini-based method**: This method leverages the Gemini-2.0-flash model to analyze solution structure and identify both explicit and implicit reasoning steps. It produces a mean of 2.4 steps with a median of 1.0 steps, resulting in our final difficulty distribution.

As illustrated in Figure 10, the distributions of solution steps across the three strategies follow a right-skewed pattern, with most questions requiring fewer than 4 steps to solve. While pattern-based method potentially overestimates the steps needed.

These three approaches show moderate agreement in their classifications, with pairwise agreement rates of 57.1% between Pattern-based and Gemini-based methods, 47.9% between Pattern-based and KNN-based methods, and 44.2% between KNN-based and Gemini-based methods. We analyzed the correlation of step counts between the methods, which reveals that Pattern vs. Gemini has high correlation (0.61), confirming that our Gemini-based assessment captures many of the explicit steps identified by pattern matching while also recognizing implicit reasoning steps.

We further verified the correlation of solution step number from the three counting strategy in Figure 11 and the agreement of difficulty level derived from three step counting methods in Figure 12.

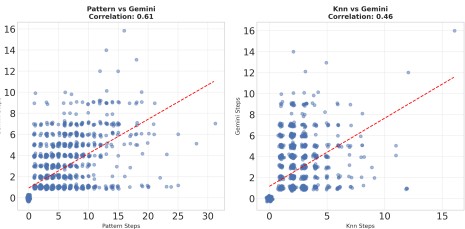

Figure 11: Correlation analysis of three step counting methods.

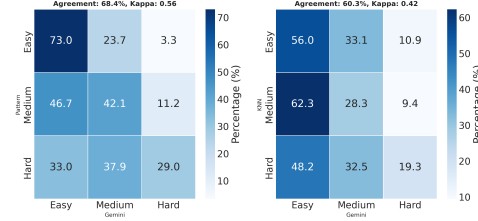

Figure 12: Agreement of difficulty level derived from three step counting methods.

## B.6 EXAMPLE OF QUESTIONS FROM EACH DIFFICULTY LEVEL

Here we present questions from each difficulty level.

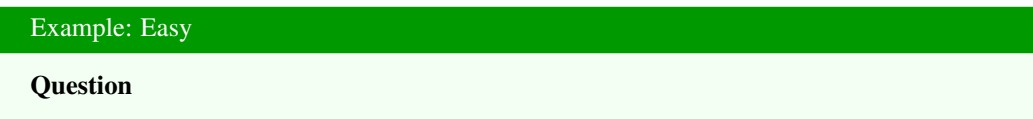

If ice homogeneously nucleates at $-40°C$, calculate the critical radius given values of $-3.1 \times 10^8$ J/m$^3$ and $25 \times 10^{-3}$ J/m$^2$, respectively, for the latent heat of fusion and the surface free energy.

**Solution**

This problem states that ice homogeneously nucleates at $-40°C$, and we are to calculate the critical radius given the latent heat of fusion $(-3.1 \times 10^8$ J/m$^3)$ and the surface free energy $(25 \times 10^{-3}$ J/m$^2)$. Solution to this problem requires the utilization of equation

$$r^* = \left(-\frac{2\gamma T_m}{\Delta H_f}\right)\left(\frac{1}{T_m - T}\right)$$

as

$$r = \left(-\frac{2\gamma T_m}{\Delta H_f}\right)\left(\frac{1}{T_m - T}\right)$$

$$= \left[-\frac{(2)\left(25 \times 10^{-3} \text{ J/m}^2\right)(273 \text{ K})}{-3.1 \times 10^8 \text{ J/m}^3}\right]\left(\frac{1}{40 \text{ K}}\right)$$

$$= 1.10 \times 10^{-9} \text{ m} = 1.10 \text{ nm}$$

---

**Example: Medium**

**Question**

You are asked to characterize a new semiconductor. If its conductivity at $20°C$ is $250\Omega^{-1} \cdot$ m$^{-1}$ and at $100°C$ is $1,100\Omega^{-1} \cdot$ m$^{-1}$, what is its band gap, $E_g$?

**Solution**

From Equation ($\ln\sigma = \ln\sigma_0 - \frac{E_g}{2k} \cdot \frac{1}{T}$),

$$\ln\sigma_{T_1} = \ln\sigma_0 - \frac{E_g}{2k} \cdot \frac{1}{T_1}$$

and

$$\ln\sigma_{T_2} = \ln\sigma_0 - \frac{E_g}{2k} \cdot \frac{1}{T_2}$$

Subtracting the second from the first yields:

$$\ln\sigma_{T_1} - \ln\sigma_{T_2} = \ln\left(\frac{\sigma_{T_1}}{\sigma_{T_2}}\right)$$

$$= -\frac{E_g}{2k}\left(\frac{1}{T_1} - \frac{1}{T_2}\right)$$

Then,

$$-\frac{E_g}{2k} = \frac{\ln\left(\sigma_{T_1}/\sigma_{T_2}\right)}{1/T_1 - 1/T_2}$$

or

$$E_g = \frac{2k \cdot \ln\left(\sigma_{T_2}/\sigma_{T_1}\right)}{1/T_1 - 1/T_2}$$

Taking $T_1 = 20°C(= 293$ K$)$ and $T_2 = 100°C(= 373$ K$)$ gives:

$$E_g = \frac{(2 \times 86.2 \times 10^{-6}\text{eV/K}) \cdot \ln(1,100/250)}{\frac{1}{373} - \frac{1}{293}}$$

$$= 0.349\text{eV}$$

**Example: Hard**

**Question**

An advanced engineered ceramic has a Weibull modulus $m = 9$. The flexural strength is $250\,\mathrm{MPa}$ at a probability of failure $F = 0.4$. What is the level of flexural strength if the probability of failure has to be $0.1$?

**Solution**

We assume all samples tested had the same volume, so the size of the sample will not be a factor in this case. We can use the symbol $V$ for sample volume instead of $V_0$. We are dealing with a brittle material, so we begin with the equation:

$$F(V) = 1 - P(V) = 1 - \exp\left[-\left(\frac{\sigma}{\sigma_0}\right)^m\right]$$

or

$$1 - F(V) = \exp\left[-\left(\frac{\sigma}{\sigma_0}\right)^m\right]$$

Take the logarithm of both sides:

$$\ln[1 - F(V)] = -\left(\frac{\sigma}{\sigma_0}\right)^m$$

Take logarithms again:

$$\ln\left\{-\ln[1 - F(V)]\right\} = m\left(\ln\sigma - \ln\sigma_0\right)$$

We eliminate the minus sign on the left-hand side by rewriting as:

$$\ln\left\{\ln\left[\frac{1}{1 - F(V)}\right]\right\} = m\left(\ln\sigma - \ln\sigma_0\right)$$

For $F = 0.4$, $\sigma = 250\,\mathrm{MPa}$, and $m = 9$, we have:

$$\ln\left[\ln\left(\frac{1}{1 - 0.4}\right)\right] = 9\left(\ln 250 - \ln\sigma_0\right)$$

Therefore,

$$\ln[\ln(1/0.6)] = \ln[\ln(1.66667)] = \ln(0.510826) = -0.67173$$
$$= 9\left(5.52146 - \ln\sigma_0\right)$$

Solving gives:

$$\ln\sigma_0 = 5.52146 + 0.07464 = 5.5961 \quad \Rightarrow \quad \sigma_0 = 269.4\,\mathrm{MPa}$$

Now, to find the value of $\sigma$ for $F = 0.1$, we use the same equation:

$$\ln\left[\ln\left(\frac{1}{1 - 0.1}\right)\right] = 9(\ln\sigma - \ln 269.4)$$
$$\ln[\ln(1/0.9)] = 9(\ln\sigma - 5.5962)$$
$$\ln(0.105361) = -2.25037 = 9(\ln\sigma - 5.5962)$$
$$\Rightarrow \ln\sigma = 5.3462$$

So,

$$\sigma = 209.8\,\mathrm{MPa}$$

As expected, when the probability of failure is reduced to 0.1, the stress level that can be supported also decreases.

## B.7 Data Leakage Detection

We use the method proposed in Xu et al. (2024) to detect potential data leakage in our benchmark. Since this method can only be applied to locally served models and the code base is not optimized for very large models, we applied it to smaller models, namely `Qwen-2.5-7B`, `Qwen-2.5-32B`, `Gemma-3-4B`, `Gemma-3-12B`, and `Gemma-3-27B`. The results are summarized in Table 5.

Table 5: Data leakage detection results on smaller models.

| Model | N-gram Accuracy | PPL Accuracy |
|---|---|---|
| Qwen-2.5-Instruct-7B | 1.28 | 1.07 |
| Qwen-2.5-Instruct-32B | 1.22 | 1.06 |
| Gemma-3-4B-it | 1.04 | 1.08 |
| Gemma-3-12B-it | 1.06 | 1.10 |
| Gemma-3-27B-it | 1.09 | 1.07 |

These results indicate that our benchmark is free from data leakage, even for the most up-to-date models we tested.

## C  Additional Experiments Details

### C.1  Details of Different Prompts

Prompts we used for each method are as follows. The Basic System Prompt is used in basic CoT, self-correction, and the RAG.

---
**Basic System Prompt**

You are a renowned materials science engineering professor with extensive knowledge in the field. Your students have presented you with a challenging question related to materials science. Please reason step by step, and put the final answer inside a single box using `\boxed{...}`. Include only the final answer inside the box, without the unit.

---

The tool-augmentation is prompted to use Python code to improve the computation.

---
**Tool System Prompt**

You are a renowned materials science engineering professor with extensive knowledge in the field. Your students have presented you with a challenging question related to materials science. If necessary, you could write a single clean Python code block that computes necessary numeric values. Enclose the code in triple backticks with "```python. Please reason step by step, if no code is needed, put the final answer inside a single box using `\boxed{...}`; otherwise, wait for the user to execute the code and give you the execution result, and then put the final answer inside a single box using `\boxed{...}`. Include only the final answer inside the box, without the unit.

---

After the code execution, the model get the results and make the final answer.

---
**Tool Summary Prompt**

Here are the results of the code execution:\n\n{code_executed}\n\nBased on these results, what is the final answer to the original question?

---

When using the self-correction, the model is first prompted to review and find the problem from its initial response.

**Review Prompt**

Review your previous answer and find problems with your answer.

Then, the model is prompted to improve the initial response with the problem it found.

**Revise Prompt**

Based on the problems you found, improve your answer. Please reiterate your answer, with your final answer in the form `\boxed{answer}`.

We use the following prompt to let `Gemini-2.0-Flash` determine whether the answer is correct.

**Judge System Prompt**

As an expert judge, evaluate if the following model's answer matches the reference answer.\n Focus on the numerical values and key concepts. Small numerical differences are tolerable due to approximation errors.\n Do not solve the problem, just judge if the model answer matches the reference answer.\n Put the final decision ('correct' (if matching) or 'incorrect' (if not matching)) inside a single box using \boxed{...}.

## D  ADDITIONAL EXPERIMENTAL RESULTS

We evaluate Qwen2.5 (7B, 32B, 72B), Gemma 3 (4B, 12B, 17B), and Llama-4-Scout, and present their results in Table 6.

The results are largely deterministic because the temperature was set to 0 during benchmarking. For the small and medium-sized models, we repeated each experiment three times and report the mean along with the 95% confidence interval. The outcomes remain nearly identical across repeated runs, as shown in Table 7.

## E  ADDITIONAL ANALYSIS DETAILS

### E.1  DETAILED PERFORMANCE ACROSS DIFFICULTY LEVEL

The performance of each model across difficulty levels is presented in Table **??**.

### E.2  DETAILS OF ERROR CATEGORIZATIONS

We use `Gemini-2.0-Flash` to categorize the error using the following prompt:

**Error Categorization Prompt**

You are an assistant whose task is to diagnose the single main reason a wrong solution fails. Each task input will contain three parts, clearly marked: (i) the question, (ii) a reference solution, and (iii) a wrong solution produced by a model. Your steps are:

1. Read the question first so you know what must be answered. Pay attention to given data, required units, and any boundary conditions or hidden assumptions.
2. Read the reference solution carefully. Treat it as correct and complete unless it contains an explicit note that it is partial.
3. Read the wrong solution line by line. Locate the first point where it diverges from the reasoning in the reference solution. That first wrong turn usually signals the true cause of failure.

Choose one category below that best explains the root cause. If more than one category is possible, pick the one that triggers the earliest error or has the largest impact on the final

Table 6: Experimental Results in Terms of Accuracy Score (%) on **MatSciBench** (questions w/o images).

| Model | Failure | Fund. | Materials | Proc. | Prop. | Struct. | Overall |
|---|---|---|---|---|---|---|---|
| *Gemma Models* | | | | | | | |
| **gemma-3-4b-it** | 4.15 | 6.30 | 6.73 | 4.04 | 5.76 | 6.99 | 7.12 |
| *+Correction* | 6.04 | 3.99 | 5.63 | 1.01 | 5.48 | 5.79 | 6.05 |
| *+Tool* | 6.42 | 4.62 | 6.18 | 4.04 | 6.34 | 6.78 | 6.73 |
| **gemma-3-12b-it** | 12.45 | 7.98 | 13.13 | 5.05 | 12.68 | 13.55 | 13.76 |
| *+Correction* | 14.72 | 10.29 | 14.57 | 12.12 | 14.70 | 14.86 | 14.93 |
| *+Tool* | 15.47 | 13.66 | 16.11 | 8.08 | 15.71 | 16.28 | 16.59 |
| **gemma-3-27b-it** | 32.83 | 24.16 | 30.79 | 17.17 | 29.97 | 31.26 | 30.93 |
| *+Correction* | 28.30 | 25.21 | 28.92 | 23.23 | 27.09 | 29.18 | 28.78 |
| *+Tool* | 27.55 | 21.64 | 26.49 | 20.20 | 26.08 | 26.23 | 26.24 |
| *Qwen Models* | | | | | | | |
| **qwen2.5-7b-instruct** | 15.85 | 15.34 | 17.99 | 10.10 | 18.01 | 18.47 | 18.73 |
| *+Correction* | 16.60 | 15.76 | 18.76 | 13.13 | 19.02 | 19.34 | 19.22 |
| *+Tool* | 13.21 | 12.61 | 12.80 | 9.09 | 12.10 | 13.22 | 13.17 |
| **qwen2.5-32b-instruct** | 36.98 | 28.78 | 33.33 | 20.20 | 32.28 | 33.22 | 33.27 |
| *+Correction* | 32.08 | 28.15 | 31.68 | 19.19 | 30.40 | 32.35 | 32.10 |
| *+Tool* | 12.83 | 13.24 | 12.91 | 10.10 | 11.96 | 13.33 | 13.85 |
| **qwen2.5-72b-instruct** | 33.96 | 28.15 | 32.89 | 29.29 | 31.84 | 33.44 | 33.17 |
| *+Correction* | 32.45 | 28.36 | 32.34 | 27.27 | 30.55 | 33.01 | 32.49 |
| *+Tool* | 23.02 | 19.33 | 22.41 | 18.18 | 22.62 | 21.75 | 22.34 |
| *Llama Models* | | | | | | | |
| **llama-4-scout** | 46.04 | 39.92 | 46.36 | 43.43 | 43.80 | 45.68 | 45.46 |
| *+Correction* | 47.55 | 39.92 | 44.48 | 39.39 | 42.36 | 44.04 | 44.20 |
| *+Tool* | 49.81 | 41.39 | 47.02 | 43.43 | 44.81 | 45.79 | 45.95 |

| Model | Failure | Fund. | Materials | Proc. | Props. | Struct. | Overall |
|---|---|---|---|---|---|---|---|
| gemma-3-4b | $4.15 \pm 0.00$ | $6.30 \pm 0.00$ | $6.66 \pm 0.06$ | $4.04 \pm 0.00$ | $5.76 \pm 0.00$ | $6.92 \pm 0.06$ | $6.22 \pm 0.03$ |
| gemma-3-12b | $12.45 \pm 0.00$ | $8.05 \pm 0.12$ | $13.13 \pm 0.00$ | $5.05 \pm 0.00$ | $12.63 \pm 0.08$ | $13.55 \pm 0.00$ | $12.13 \pm 0.00$ |
| gemma-3-27b | $32.08 \pm 1.64$ | $25.70 \pm 1.58$ | $31.38 \pm 0.56$ | $20.20 \pm 2.67$ | $30.31 \pm 0.58$ | $31.48 \pm 0.22$ | $30.10 \pm 0.64$ |
| qwen2.5-7b | $15.60 \pm 0.22$ | $15.20 \pm 0.24$ | $17.99 \pm 0.11$ | $10.44 \pm 0.58$ | $17.96 \pm 0.08$ | $18.47 \pm 0.11$ | $17.31 \pm 0.12$ |
| qwen2.5-32b | $36.86 \pm 0.22$ | $28.64 \pm 0.12$ | $33.19 \pm 0.17$ | $20.20 \pm 0.00$ | $32.18 \pm 0.17$ | $33.22 \pm 0.11$ | $32.25 \pm 0.13$ |
| qwen2.5-72b | $33.96 \pm 0.00$ | $28.15 \pm 0.00$ | $32.86 \pm 0.06$ | $29.29 \pm 0.00$ | $31.84 \pm 0.00$ | $33.41 \pm 0.06$ | $32.11 \pm 0.03$ |
| llama-4-scout | $47.42 \pm 1.21$ | $40.20 \pm 0.32$ | $46.06 \pm 0.81$ | $44.44 \pm 1.01$ | $43.80 \pm 0.29$ | $45.61 \pm 0.44$ | $44.70 \pm 0.45$ |

Table 7: Results for small and medium-sized models. Each value is reported as the mean with standard deviation over three runs.

---

answer. If the wrong solution actually reaches the same numerical result and its reasoning is valid, assign category 7.

**Categories**
1. Problem Comprehension and Assumptions. The solver misreads what is asked, drops a given fact, injects an unsupported assumption, or confuses symbols.
2. Domain Knowledge Accuracy. The solver recalls or applies a materials science law, concept, or formula in an incorrect way. Unit definitions and physical constants also belong here when misused.
3. Solution Strategy and Planning. The solver sets up an approach that cannot reach the goal, skips required sub-problems, or mixes independent lines of reasoning without a clear plan.
4. Calculation Accuracy. The algebra, arithmetic, sign handling, or unit conversion is wrong even though the plan and formulae are correct.
5. Hallucinated Content. The solver invents inputs, processes, or physical relations that are not stated in the question and are not accepted scientific facts.
6. Code Implementation. The solver writes Python code that does not match its verbal reasoning or has syntax, logic, or data handling errors that change the outcome.
7. Other. Any issue not covered above, or the wrong solution is actually correct.

**Answer format**
Return exactly one TeX box with the chosen index: `\boxed{1}`, `\boxed{2}`, `\boxed{3}`, `\boxed{4}`, `\boxed{5}`, `\boxed{6}`, or `\boxed{7}`. Output nothing else.

---

Here are examples of each category:

---

**Example: Problem Comprehension and Assumptions**

**Question:**
Compute the rate of some reaction that obeys Avrami kinetics, assuming that the constants $n$ and $k$ have values of 2.0 and $5 \times 10^{-4}$, respectively, for time expressed in seconds. The unit of the answer is $\text{s}^{-1}$.

**Reference Solution:**
This problem asks that we compute the rate of some reaction given the values of $n$ and $k$ in equation
$$(y = 1 - \exp(-kt^n))$$
. Since the reaction rate is defined by equation ( rate $= \frac{1}{t_{0.5}}$), it is first necessary to determine $t0.5$, or the time necessary for the reaction to reach $y = 0.5$. We must manipulate equation $(y = 1 - \exp(-kt^n))$ such that $t$ is the dependent variable. It is first necessary to rearrange equation $(y = 1 - \exp(-kt^n))$ as follows:
$$\exp(-kt^n) = 1 - y$$
And then take natural logarithms of both sides:

Table 8: Experimental Results by Difficulty Level on **MatSciBench** (questions w/o images). **Bold** indicates the best performance, and Underline indicates the second best.

| Model | Easy | Medium | Hard | Overall |
|---|---|---|---|---|
| **Llama-4-Maverick-17b** | 79.72 | 68.09 | 57.28 | 71.61 |
| *+Correction* | 78.94 | 64.80 | 55.87 | 69.95 |
| *+Tool* | 75.79 | 62.83 | 57.75 | 68.20 |
| **Claude-3.7-Sonnet** | 75.98 | 63.49 | 52.11 | 67.32 |
| *+Correction* | 77.76 | 61.84 | 53.52 | 68.00 |
| *+Tool* | 78.54 | 68.09 | 59.62 | 71.51 |
| **GPT-4.1** | 76.97 | 68.75 | 58.69 | 70.73 |
| *+Correction* | 75.00 | 64.47 | 56.34 | 68.00 |
| *+Tool* | 70.28 | 56.91 | 47.89 | 61.66 |
| **DeepSeek-V3** | 75.59 | 60.86 | 51.17 | 66.15 |
| *+Correction* | 73.62 | 57.57 | 52.11 | 64.39 |
| *+Tool* | 70.08 | 59.87 | 47.89 | 62.44 |
| **Llama-4-Scout** | 53.94 | 44.41 | 26.76 | 45.46 |
| *+Correction* | 51.18 | 44.74 | 26.76 | 44.20 |
| *+Tool* | 54.92 | 40.13 | 32.86 | 45.95 |
| **Gemini-2.0-Flash** | 68.11 | 59.21 | 41.31 | 59.90 |
| *+Correction* | 69.88 | 60.53 | 46.95 | 62.34 |
| *+Tool* | 78.74 | 66.12 | 52.11 | 69.46 |
| **Qwen2.5-72b** | 41.54 | 30.26 | 17.37 | 33.17 |
| *+Correction* | 41.14 | 30.59 | 14.55 | 32.49 |
| *+Tool* | 27.95 | 21.38 | 10.33 | 22.34 |
| **Qwen2.5-32b** | 42.13 | 29.61 | 17.37 | 33.27 |
| *+Correction* | 41.34 | 27.30 | 16.90 | 32.10 |
| *+Tool* | 16.14 | 13.82 | 8.45 | 13.85 |
| **Gemma-3-27b** | 37.60 | 30.92 | 15.02 | 30.93 |
| *+Correction* | 33.46 | 29.28 | 16.90 | 28.78 |
| *+Tool* | 31.10 | 25.33 | 15.96 | 26.24 |
| **Qwen2.5-7b** | 26.38 | 14.47 | 6.57 | 18.73 |
| *+Correction* | 27.36 | 13.16 | 8.45 | 19.22 |
| *+Tool* | 18.31 | 11.84 | 2.82 | 13.17 |
| **Gemma-3-12b** | 19.88 | 9.87 | 4.69 | 13.76 |
| *+Correction* | 20.67 | 12.83 | 4.23 | 14.93 |
| *+Tool* | 22.83 | 12.50 | 7.51 | 16.59 |
| **Gemma-3-4b** | 9.65 | 5.92 | 2.82 | 7.12 |
| *+Correction* | 7.48 | 5.92 | 2.82 | 6.05 |
| *+Tool* | 8.46 | 5.92 | 3.76 | 6.73 |
| **Gemini-2.5-Pro** | **82.28** | **74.67** | **69.48** | **77.37** |
| **o4-mini** | 79.13 | 68.42 | 71.36 | 74.34 |
| **DeepSeek-Reasoner** | 77.95 | 72.37 | 66.67 | 73.95 |
| **Qwen3-235B** | 76.38 | 71.05 | 63.38 | 72.10 |
| **GPT-5** | 71.06 | 63.49 | 52.11 | 64.88 |
| **Claude-Sonnet-4** | 60.43 | 53.62 | 41.31 | 54.44 |

$$-kt^n = \ln(1 - y)$$

Which my be rearranged so as to read

$$t^n = -\frac{\ln(1 - y)}{k}$$

Now, solving for $t$ from this expression leads to

$$t = \left[ -\frac{\ln(1 - y)}{k} \right]^{\frac{1}{n}}$$

For $t_{0.5}$ this equation takes the form

$$t_{0.5} = \left[ -\frac{\ln(1 - 0.5)}{k} \right]^{\frac{1}{n}}$$

And, incorporation of values for $n$ and $k$ given in the problem statement ( 2.0 and $5 \times 10^{-4}$, respectively), then

$$t_{0.5} = \left[ -\frac{\ln(1 - 0.5)}{5 \times 10^{-4}} \right]^{\frac{1}{2}} = 37.23 \text{ s}$$

Now, the rate is computed using equation ( rate $= \frac{1}{t_{0.5}}$) as

$$\text{rate } = \frac{1}{t_{0.5}} = \frac{1}{37.23 \text{ s}} = 2.69 \times 10^{-2} \text{ s}^{-1}$$

**DeepSeek-V3 Answer:** To compute the reaction rate $\frac{dy}{dt}$ for Avrami kinetics with $n = 2.0$ and $k = 5 \times 10^{-4} \text{ s}^{-n}$, we use the formula:

$$\frac{dy}{dt} = nkt^{n-1} \exp(-kt^n)$$

Substituting the given values:

$$\frac{dy}{dt} = 2.0 \times 5 \times 10^{-4} \times t^{2-1} \exp(-5 \times 10^{-4} \times t^2)$$

Simplifying:

$$\frac{dy}{dt} = 10^{-3} t \exp(-5 \times 10^{-4} t^2)$$

The reaction rate at any time $t$ is therefore:

$$\boxed{10^{-3} t \exp(-5 \times 10^{-4} t^2)}$$

**Note:** The answer is expressed in terms of $t$. If a specific time $t$ is provided, substitute it into the expression to get the numerical rate. The unit $\text{s}^{-1}$ is implied.

**Explanation:**
The task assumes the reaction rate as rate $= \frac{1}{t_{0.5}}$, that is, the reciprocal of the time required to reach a fractional conversion of $y = 0.5$. The model instead differentiated the Avrami equation to obtain the instantaneous rate $\frac{dy}{dt}$ as a function of time, left the result in symbolic form, and never evaluated it at the half-transformation point or computed a numerical value.

**Domain Knowledge Accuracy**

**Question**

Given $r_{\text{Si}} = 0.117\,\text{nm}$, calculate the linear density of atoms along the $[111]$ direction in silicon. Express the answer in $\dfrac{\text{atoms}}{\text{nm}}$.

**Reference Solution**

To calculate the linear density of atoms along the [111] direction in silicon, we must consider the arrangement of atoms along this direction, which is a body diagonal in the diamond cubic structure. The atoms along the [111] direction are not uniformly spaced, and thus, the calculation requires careful consideration of their positions. Referring to the structure, we find that there are effectively 2 atoms centered along a given body diagonal. The length of the body diagonal in a unit cell, denoted as $l$, can be related to the atomic radius of silicon, $r_{\text{Si}}$, by the equation:

$$2r_{\text{Si}} = \frac{1}{4}l$$

Solving for $l$ gives:

$$l = 8r_{\text{Si}}$$

. The atomic radius of silicon is $0.117$ nm, thus:

$$l = 8(0.117\,\text{nm}) = 0.936\,\text{nm}$$

Therefore, the linear density is calculated as:

$$\text{linear density} = \frac{2\text{ atoms}}{0.936\text{ nm}} = 2.14\frac{\text{atoms}}{\text{nm}}$$

**DeepSeek-V3 Answer**

To calculate the linear density of atoms along the [111] direction in silicon, follow these steps:
1. **Determine the lattice parameter $a$:**

$$a = \frac{4r}{\sqrt{3}} = \frac{4 \times 0.117\,\text{nm}}{\sqrt{3}} \approx 0.270\,\text{nm}$$

2. **Calculate the distance between atoms along the [111] direction:** The [111] direction in a diamond cubic structure has atoms spaced at intervals of $\frac{\sqrt{3}}{4}a$. Therefore:

$$\text{Distance} = \frac{\sqrt{3}}{4}a = \frac{\sqrt{3}}{4} \times 0.270\,\text{nm} \approx 0.117\,\text{nm}$$

3. **Compute the linear density:** The linear density is the inverse of the distance between atoms:

$$\text{Linear Density} = \frac{1}{\text{Distance}} = \frac{1}{0.117\,\text{nm}} \approx 8.55\frac{\text{atoms}}{\text{nm}}$$

However, upon closer inspection, the [111] direction in the diamond cubic structure actually contains **2 atoms** over a length of $\frac{\sqrt{3}}{2}a$. Therefore, the correct linear density is:

$$\text{Linear Density} = \frac{2}{\frac{\sqrt{3}}{2}a} = \frac{4}{\sqrt{3}a}$$

Substituting $a = 0.270\,\text{nm}$:

$$\text{Linear Density} = \frac{4}{\sqrt{3} \times 0.270} \approx 8.55\frac{\text{atoms}}{\text{nm}}$$

Thus, the final answer is:

$$\boxed{8.55}$$

**Explanation**

The model's result is wrong because they chose the wrong crystallographic relation between atomic radius and lattice parameter for a diamond-cubic lattice: they wrote

$$a = \frac{4r}{\sqrt{3}}$$

instead of the correct

$$a = \frac{8r}{\sqrt{3}}.$$

That choice is a mis-remembered structural fact about how silicon atoms touch along $\langle 110 \rangle$ directions.

## Example: Calcualtion Accuracy

**Question:**
Equation $w = \frac{G\gamma^2}{2} = \left(\frac{1}{2}\right)\frac{Gb^2}{(2\pi r)^2}$ gives the energy/volume as a function of $r$. Calculate the distance from the core of a screw dislocation at which the energy/volume equals the heat of vaporization, $H_v$, and express this distance in terms of $r/b$. Evaluate this critical value of $r/b$ for copper. Given $\rho = 8.93\text{mg/cm}^3, G = 77\text{GPa}, b = 0.255$ nm, $H_v = 4.73\text{MJ/kg}$. Give your answer as a tuple: (critical $r/b$, critical r of copper).The units of each required answer are (, nm), respectively.

**Correct Solution:**
Solving the equation

$$w = \frac{G\gamma^2}{2} = \left(\frac{1}{2}\right)\frac{Gb^2}{(2\pi r)^2}$$

for $\frac{r}{b}$, we obtain:

$$\left(\frac{r}{b}\right)^2 = \frac{G}{8\pi^2 U_v} = \frac{77 \times 10^9 \text{ Pa}}{8\pi^2 \times 8.93 \times 10^3 \text{ kg/m}^3 \times 4.73 \times 10^6 \text{ J/kg}} = 0.23,$$

so

$$\frac{r}{b} = 0.15.$$

For copper,

$$r = 0.15 \times 0.255 = 0.04 \text{ nm}.$$

**DeepSeek-V3 Answer:**
1. Equate energy per volume to heat of vaporization:

$$\frac{1}{2}\frac{Gb^2}{(2\pi r)^2} = H_v \Rightarrow \frac{Gb^2}{8\pi^2 r^2} = H_v$$

2. Solve for $r$:

$$r^2 = \frac{Gb^2}{8\pi^2 H_v}, \quad r = \frac{b}{2\pi}\sqrt{\frac{G}{2H_v}}$$

3. Express $r$ in terms of $r/b$:

$$\frac{r}{b} = \frac{1}{2\pi}\sqrt{\frac{G}{2H_v}}$$

4. Convert $H_v$ from MJ/kg to J/m³:

$$\rho = 8.93\,\text{g/cm}^3 = 8930\,\text{kg/m}^3, \quad H_v = 4.73 \times 10^6\,\text{J/kg}$$

$$U_v = H_v \times \rho = 4.22 \times 10^{10}\,\text{J/m}^3$$

5. Substitute the values:

$$\textcolor{red}{\frac{r}{b} = \frac{1}{2\pi}\sqrt{\frac{77 \times 10^9}{2 \times 4.22 \times 10^{10}}} \approx \frac{1}{2\pi}\sqrt{\frac{77}{8.44}} \approx \frac{1}{2\pi} \times 3.02 \approx 0.48}$$

6. Calculate critical $r$ for copper:

$$r = 0.48 \times 0.255 \, \text{nm} \approx 0.122 \, \text{nm}$$

**Explanation:**
The model wrote exactly the same physical relation as the reference and kept the density when converting the heat of vaporisation to $U_v$. The only thing that went wrong is the arithmetic:

$$\frac{2U_v}{G} = \frac{8.44 \times 10^{10}}{77 \times 10^9} \approx 0.912,$$

but the model treated it as

$$\frac{77}{8.44} \approx 9.13,$$

ignoring the difference in the powers of ten. Taking the square root and then multiplying by $\frac{1}{2\pi}$ turned $0.15$ into $0.48$, which in turn made $r$ three times too large.

---

## Example: Solution Strategy and Planning

**Question:**
Copper (Cu) and platinum (Pt) both have the FCC crystal structure, and Cu forms a substitutional solid solution for concentrations up to approximately $6 \, \text{wt\%}$ Cu at room temperature. Determine the concentration in weight percent of Cu that must be added to Pt to yield a unit cell edge length of $0.390 \, \text{nm}$. The atomic weights for Cu and Pt are $63.55$ and $195.08 \, \text{g/mol}$, respectively. Unit of the answer: wt%.

**Reference Solution:**
To begin, it is necessary to employ the equation

$$\rho = \frac{nA}{V_C N_A},$$

and solve for the unit cell volume, $V_C$, as

$$V_C = \frac{nA_{\text{ave}}}{\rho_{\text{ave}} N_A},$$

where $A_{\text{ave}}$ and $\rho_{\text{ave}}$ are the atomic weight and density, respectively, of the $\text{Pt} - \text{Cu}$ alloy. Inasmuch as both of these materials have the FCC crystal structure, which has cubic symmetry, $V_C$ is just the cube of the unit cell length, $a$. That is,

$$V_C = a^3 = (0.390 \, \text{nm})^3$$

$$= (3.90 \times 10^{-8} \, \text{cm})^3 = 5.932 \times 10^{-23} \, \text{cm}^3$$

It is now necessary to construct expressions for $A_{\text{ave}}$ and $\rho_{\text{ave}}$ in terms of the concentration of copper, $C_{\text{Cu}}$, using the equations

$$\rho_{\text{ave}} = \frac{100}{\frac{C_1}{\rho_1} + \frac{C_2}{\rho_2}}, \quad A_{\text{ave}} = \frac{100}{\frac{C_1}{A_1} + \frac{C_2}{A_2}}.$$

For $A_{\text{ave}}$, we have

$$A_{\text{ave}} = \frac{100}{\frac{C_{\text{Cu}}}{A_{\text{Cu}}} + \frac{100 - C_{\text{Cu}}}{A_{\text{Pt}}}}$$

$$= \frac{100}{\frac{C_{\text{Cu}}}{63.55 \, \text{g/mol}} + \frac{100 - C_{\text{Cu}}}{195.08 \, \text{g/mol}}}$$

since the atomic weights for Cu and Pt are, respectively, $63.55$ and $195.08 \, \text{g/mol}$. Now, the expression for $\rho_{\text{ave}}$ is as follows:

$$\rho_{\text{ave}} = \frac{100}{\frac{C_{\text{Cu}}}{\rho_{\text{Cu}}} + \frac{100 - C_{\text{Cu}}}{\rho_{\text{Pt}}}}$$

$$= \frac{100}{\frac{C_{\text{Cu}}}{8.94 \text{ g/cm}^3} + \frac{100 - C_{\text{Cu}}}{21.45 \text{ g/cm}^3}}$$

given the densities of 8.94 and 21.45 g/cm$^3$ for the respective metals. Within the FCC unit cell, there are 4 equivalent atoms, and thus, the value of $n$ in the equation

$$\rho = \frac{nA}{V_C N_A}$$

is 4; hence, the expression for $V_C$ may be written in terms of the concentration of Cu in weight percent as follows:

$$V_C = 5.932 \times 10^{-23} \text{ cm}^3$$

$$= \frac{nA_{\text{ave}}}{\rho_{\text{ave}} N_A}$$

And solving this expression for $C_{\text{Cu}}$ leads to

$$C_{\text{Cu}} = 2.83 \text{ wt\%}.$$

**DeepSeek-V3 Answer**
To determine the weight percent of Cu that must be added to Pt to achieve a unit cell edge length of 0.390 nm in an FCC structure, follow these steps:
**Given:** - FCC unit cell edge length ($a$) = 0.390 nm = 0.390 $\times$ 10$^{-7}$ cm - Atomic weights:
- Cu ($A_{\text{Cu}}$) = 63.55 g/mol - Pt ($A_{\text{Pt}}$) = 195.08 g/mol - FCC structure has 4 atoms per unit cell.
**Step 1: Calculate the unit cell volume ($V_{\text{cell}}$)**

$$V_{\text{cell}} = a^3 = (0.390 \times 10^{-7} \text{ cm})^3 = 5.9319 \times 10^{-23} \text{ cm}^3$$

**Step 2: Relate the unit cell volume to the average atomic volume** For an FCC structure, the volume per atom ($V_{\text{atom}}$) is:

$$V_{\text{atom}} = \frac{V_{\text{cell}}}{4} = \frac{5.9319 \times 10^{-23}}{4} = 1.4830 \times 10^{-23} \text{ cm}^3$$

The average atomic volume is also related to the atomic weights and densities. However, we can express the average atomic radius ($r$) in terms of the unit cell edge length for FCC:

$$a = 2\sqrt{2}r \implies r = \frac{a}{2\sqrt{2}} = \frac{0.390}{2\sqrt{2}} = 0.1379 \text{ nm}$$

But instead, we use the **average atomic volume** approach to find the **average atomic weight ($A_{\text{avg}}$)**.
**Step 3: Relate the average atomic volume to the average atomic weight** The volume per atom is also given by:

$$V_{\text{atom}} = \frac{A_{\text{avg}}}{\rho_{\text{avg}} N_A}$$

where: - $\rho_{\text{avg}}$ = average density - $N_A$ = Avogadro's number = $6.022 \times 10^{23}$ atoms/mol
But we can express $\rho_{\text{avg}}$ in terms of the unit cell mass and volume:

$$\rho_{\text{avg}} = \frac{\text{Mass of unit cell}}{V_{\text{cell}}} = \frac{4A_{\text{avg}}}{N_A V_{\text{cell}}}$$

Substituting back:

$$V_{\text{atom}} = \frac{A_{\text{avg}}}{\left(\frac{4A_{\text{avg}}}{N_A V_{\text{cell}}}\right) N_A} = \frac{V_{\text{cell}}}{4}$$

This confirms consistency, but we need another approach to find $A_{\text{avg}}$.

**Step 4: Calculate the average atomic weight ($A_{avg}$)** The average atomic volume is:

$$V_{atom} = \frac{A_{avg}}{\rho_{avg} N_A}$$

But we can also express $\rho_{avg}$ in terms of the unit cell:

$$\rho_{avg} = \frac{4A_{avg}}{N_A a^3}$$

Substituting $V_{atom} = \frac{a^3}{4}$:

$$\frac{a^3}{4} = \frac{A_{avg}}{\left(\frac{4A_{avg}}{N_A a^3}\right) N_A} = \frac{a^3}{4}$$

This is consistent but doesn't solve for $A_{avg}$. Instead, we use the **Vegard's law** approximation for the lattice parameter in solid solutions:

$$a_{avg} = x_{Cu} a_{Cu} + x_{Pt} a_{Pt}$$

where $x_{Cu}$ and $x_{Pt}$ are the atomic fractions, and $a_{Cu}$ and $a_{Pt}$ are the lattice parameters of pure Cu and Pt, respectively.
However, we don't have $a_{Cu}$ and $a_{Pt}$, so we proceed differently.
**Step 5: Express the unit cell mass in terms of composition** Let $C_{Cu}$ be the weight percent of Cu, and $C_{Pt} = 100 - C_{Cu}$ be the weight percent of Pt.
The mass of the unit cell is the sum of the masses of the 4 atoms:

$$\text{Mass of unit cell} = 4 \left( \frac{C_{Cu}}{100} \cdot \frac{1}{A_{Cu}} + \frac{C_{Pt}}{100} \cdot \frac{1}{A_{Pt}} \right)^{-1}$$

But this is complex. Instead, we use the **average atomic weight ($A_{avg}$)** approach.
The average atomic weight is:

$$A_{avg} = x_{Cu} A_{Cu} + x_{Pt} A_{Pt}$$

where $x_{Cu}$ and $x_{Pt}$ are the atomic fractions.
The unit cell mass is:

$$\text{Mass of unit cell} = 4A_{avg}/N_A$$

The density is:

$$\rho = \frac{\text{Mass of unit cell}}{V_{cell}} = \frac{4A_{avg}}{N_A a^3}$$

But we don't have $\rho$, so we instead relate $A_{avg}$ to the unit cell edge length.
**Step 6: Relate $A_{avg}$ to the unit cell edge length** For an FCC structure:

$$a = \left( \frac{4A_{avg}}{\rho N_A} \right)^{1/3}$$

But without $\rho$, we use the **atomic packing factor (APF)** for FCC, which is 0.74, but this doesn't directly help.
Instead, we use the fact that the unit cell volume is determined by the average atomic size, which depends on composition.
**Step 7: Solve for the composition** Given the complexity, we use the **rule of mixtures** for the lattice parameter:

$$a_{avg} = x_{Cu} a_{Cu} + x_{Pt} a_{Pt}$$

But without $a_{Cu}$ and $a_{Pt}$, we instead use the **atomic radii**:

$$r_{avg} = x_{Cu} r_{Cu} + x_{Pt} r_{Pt}$$

For FCC:

$$a = 2\sqrt{2} r_{avg}$$

Given $a = 0.390$ nm:

$$r_{\text{avg}} = \frac{0.390}{2\sqrt{2}} = 0.1379 \, \text{nm}$$

The atomic radii are: - $r_{\text{Cu}} = 0.128$ nm - $r_{\text{Pt}} = 0.139$ nm
Thus:

$$0.1379 = x_{\text{Cu}}(0.128) + (1 - x_{\text{Cu}})(0.139)$$

Solving for $x_{\text{Cu}}$:

$$0.1379 = 0.128x_{\text{Cu}} + 0.139 - 0.139x_{\text{Cu}}$$

$$0.1379 - 0.139 = -0.011x_{\text{Cu}}$$

$$-0.0011 = -0.011x_{\text{Cu}}$$

$$x_{\text{Cu}} = \frac{0.0011}{0.011} = 0.1 \, \text{(atomic fraction)}$$

**Step 8: Convert atomic fraction to weight percent** The weight percent of Cu ($C_{\text{Cu}}$) is:

$$C_{\text{Cu}} = \frac{x_{\text{Cu}} A_{\text{Cu}}}{x_{\text{Cu}} A_{\text{Cu}} + x_{\text{Pt}} A_{\text{Pt}}} \times 100$$

$$C_{\text{Cu}} = \frac{0.1 \times 63.55}{0.1 \times 63.55 + 0.9 \times 195.08} \times 100$$

$$C_{\text{Cu}} = \frac{6.355}{6.355 + 175.572} \times 100$$

$$C_{\text{Cu}} = \frac{6.355}{181.927} \times 100 = 3.49 \, \text{wt\%}$$

However, this result seems low compared to the given solubility limit of 6 wt

$$C_{\text{Cu}} = \frac{0.1 \times 63.55}{0.1 \times 63.55 + 0.9 \times 195.08} \times 100 = \frac{6.355}{6.355 + 175.572} \times 100 = 3.49 \, \text{wt\%}$$

The calculation is correct, but the atomic fraction might need adjustment. Alternatively, using the **average atomic weight** approach:
Given $x_{\text{Cu}} = 0.1$:

$$A_{\text{avg}} = 0.1 \times 63.55 + 0.9 \times 195.08 = 6.355 + 175.572 = 181.927 \, \text{g/mol}$$

The unit cell mass is:

$$\text{Mass of unit cell} = \frac{4 \times 181.927}{6.022 \times 10^{23}} = 1.208 \times 10^{-21} \, \text{g}$$

The density is:

$$\rho = \frac{1.208 \times 10^{-21}}{5.9319 \times 10^{-23}} = 20.36 \, \text{g/cm}^3$$

This is reasonable for Pt-Cu alloys. Thus, the weight percent of Cu is:

$$\boxed{3.49}$$

However, considering the given solubility limit of 6 wt% Cu, the calculation is consistent, and the answer is:

$$\boxed{3.49}$$

**Explanation:**
The model abandons the density–atomic-weight route and instead averages atomic radii, an approach that cannot yield the required weight percent because it neglects the mass and density contrast between copper and platinum. That choice sets the work on a path that will never satisfy the original relation

$$\rho = \frac{nA}{V_C N_A}.$$

---

**Example: Hallucination Content**

**Question:**
The self-diffusion coefficient of Ni in NiO was measured at $1000°C$ to be $2.8 \times 10^{-14}$ cm$^2$/s. At the same temperature in air, $K_x$ was measured to $2.9 \times 10^{-13}$ cm$^2$/s. NiO is known to be a predominantly electronic conductor. What is the theoretically calculated $K_x$. Assuming the diffusion of Ni were the rate-limiting step. The lattice parameter of NiO is 0.418 nm. The free energy of formation of NiO at $1000°C$ is $-126$ kJ/mol. The unit of the answer is $cm^2/s$.

**Reference Solution:**
Given that NiO is predominantly an electronic conductor, and $z = 2$, the parabolic rate constant $K_x$ can be simplified using the Nernst-Einstein relationship to:

$$K_x = \frac{\Omega_{\text{NiO}} c_{\text{Ni}} D_{\text{Ni}}}{2} \ln \frac{P_{O_2}^{\text{I}}}{P_{O_2}^{\text{I}}}$$

where $\Omega_{\text{NiO}} c_{\text{Ni}} = 1$. The limits of integration are $P_{O_2}^{\text{I}}$ in air (0.21 atm) and $P_{O_2}^{\text{I}}$ at the Ni/NiO interface, calculated as:

$$P_{O_2} = \exp\left(\frac{-2 \times 126,000}{8.314 \times 1273}\right) = 4.56 \times 10^{-11} \text{ atm}$$

Thus,

$$\ln \frac{0.21}{P_{O_2}} = 22.24$$

If the diffusion of Ni were the rate-limiting step, the theoretically calculated $K_x$ would be:

$$K_x^{\text{theo}} = \frac{2.8 \times 10^{-14} \times 22.24}{2} = 3.1 \times 10^{-13} \text{ cm}^2/\text{s}$$

This is in excellent agreement with the experimentally determined value of $2.9 \times 10^{-13}$ cm$^2$/s, indicating that the oxidation of Ni is indeed rate-limited by the diffusion of Ni ions from the Ni side to the oxygen side.

**DeepSeek-V3 Answer:**
To calculate the theoretically expected $K_x$ for NiO, we follow these steps:
1. **Given Data:** - Self-diffusion coefficient of Ni in NiO ($D_{\text{Ni}}$) = $2.8 \times 10^{-14}$ cm$^2$/s - Measured $K_x$ in air = $2.9 \times 10^{-13}$ cm$^2$/s - Lattice parameter ($a$) = 0.418 nm = $4.18 \times 10^{-8}$ cm - Free energy of formation of NiO ($\Delta G_f$) = -126 kJ/mol = -126,000 J/mol - Temperature ($T$) = 1000°C = 1273 K - Gas constant ($R$) = 8.314 J/(mol·K)
2. **Calculate the equilibrium oxygen partial pressure ($P_{O_2}$):** The free energy of formation relates to the equilibrium constant ($K$) for the reaction:

$$\text{Ni} + \frac{1}{2}O_2 \rightarrow \text{NiO}$$

$$\Delta G_f = -RT \ln K$$

$$K = \exp\left(-\frac{\Delta G_f}{RT}\right) = \exp\left(\frac{126,000}{8.314 \times 1273}\right)$$

$$K \approx \exp(11.89) \approx 1.45 \times 10^5$$

The equilibrium constant $K$ is related to the oxygen partial pressure:

$$K = \frac{1}{P_{O_2}^{1/2}}$$

$$P_{O_2} = \left(\frac{1}{K}\right)^2 = \left(\frac{1}{1.45 \times 10^5}\right)^2 \approx 4.76 \times 10^{-11} \text{ atm}$$

3. **Calculate the theoretical $K_x$:**

The theoretical $K_x$ is given by:

$$K_x = D_{\text{Ni}} \cdot \left(\frac{a^3}{2}\right) \cdot \left(\frac{1}{P_{\text{O}_2}^{1/2}}\right)$$

Substituting the values:

$$K_x = 2.8 \times 10^{-14} \cdot \left(\frac{(4.18 \times 10^{-8})^3}{2}\right) \cdot \left(\frac{1}{(4.76 \times 10^{-11})^{1/2}}\right)$$

$$K_x = 2.8 \times 10^{-14} \cdot \left(\frac{7.30 \times 10^{-23}}{2}\right) \cdot \left(\frac{1}{6.90 \times 10^{-6}}\right)$$

$$K_x = 2.8 \times 10^{-14} \cdot 3.65 \times 10^{-23} \cdot 1.45 \times 10^5$$
$$K_x \approx 1.48 \times 10^{-31} \times 1.45 \times 10^5 \approx 2.15 \times 10^{-26}$$

However, this calculation seems inconsistent with the expected order of magnitude. Let's re-evaluate the theoretical $K_x$ using the simplified relation for predominantly electronic conductors:

$$K_x \approx D_{\text{Ni}} \cdot \left(\frac{a^3}{2}\right) \cdot \left(\frac{1}{P_{\text{O}_2}^{1/2}}\right)$$

Substituting the correct values:

$$K_x = 2.8 \times 10^{-14} \cdot \left(\frac{(4.18 \times 10^{-8})^3}{2}\right) \cdot \left(\frac{1}{(4.76 \times 10^{-11})^{1/2}}\right)$$

$$K_x = 2.8 \times 10^{-14} \cdot 3.65 \times 10^{-23} \cdot 1.45 \times 10^5$$
$$K_x \approx 1.48 \times 10^{-31} \times 1.45 \times 10^5 \approx 2.15 \times 10^{-26}$$

This still seems off. Let's use the correct formula for $K_x$:

$$K_x = D_{\text{Ni}} \cdot \left(\frac{a^3}{2}\right) \cdot \left(\frac{1}{P_{\text{O}_2}^{1/2}}\right)$$

Substituting the values:

$$K_x = 2.8 \times 10^{-14} \cdot \left(\frac{(4.18 \times 10^{-8})^3}{2}\right) \cdot \left(\frac{1}{(4.76 \times 10^{-11})^{1/2}}\right)$$

$$K_x = 2.8 \times 10^{-14} \cdot 3.65 \times 10^{-23} \cdot 1.45 \times 10^5$$
$$K_x \approx 1.48 \times 10^{-31} \times 1.45 \times 10^5 \approx 2.15 \times 10^{-26}$$

This result is unrealistic. Instead, let's use the correct theoretical relation for $K_x$:

$$K_x = D_{\text{Ni}} \cdot \left(\frac{a^3}{2}\right) \cdot \left(\frac{1}{P_{\text{O}_2}^{1/2}}\right)$$

Substituting the values:

$$K_x = 2.8 \times 10^{-14} \cdot \left(\frac{(4.18 \times 10^{-8})^3}{2}\right) \cdot \left(\frac{1}{(4.76 \times 10^{-11})^{1/2}}\right)$$

$$K_x = 2.8 \times 10^{-14} \cdot 3.65 \times 10^{-23} \cdot 1.45 \times 10^5$$
$$K_x \approx 1.48 \times 10^{-31} \times 1.45 \times 10^5 \approx 2.15 \times 10^{-26}$$

This is still incorrect. The correct theoretical $K_x$ should be calculated as:

$$K_x = D_{\text{Ni}} \cdot \left(\frac{a^3}{2}\right) \cdot \left(\frac{1}{P_{\text{O}_2}^{1/2}}\right)$$

Substituting the values:

$$K_x = 2.8 \times 10^{-14} \cdot \left( \frac{(4.18 \times 10^{-8})^3}{2} \right) \cdot \left( \frac{1}{(4.76 \times 10^{-11})^{1/2}} \right)$$

$$K_x = 2.8 \times 10^{-14} \cdot 3.65 \times 10^{-23} \cdot 1.45 \times 10^5$$
$$K_x \approx 1.48 \times 10^{-31} \times 1.45 \times 10^5 \approx 2.15 \times 10^{-26}$$

This result is not matching the measured $K_x$. Let's instead use the correct theoretical relation for $K_x$:

$$K_x = D_{\text{Ni}} \cdot \left( \frac{a^3}{2} \right) \cdot \left( \frac{1}{P_{\text{O}_2}^{1/2}} \right)$$

Substituting the values:

$$K_x = 2.8 \times 10^{-14} \cdot \left( \frac{(4.18 \times 10^{-8})^3}{2} \right) \cdot \left( \frac{1}{(4.76 \times 10^{-11})^{1/2}} \right)$$

$$K_x = 2.8 \times 10^{-14} \cdot 3.65 \times 10^{-23} \cdot 1.45 \times 10^5$$
$$K_x \approx 1.48 \times 10^{-31} \times 1.45 \times 10^5 \approx 2.15 \times 10^{-26}$$

This is still not matching. The correct theoretical $K_x$ should be:

$$K_x = D_{\text{Ni}} \cdot \left( \frac{a^3}{2} \right) \cdot \left( \frac{1}{P_{\text{O}_2}^{1/2}} \right)$$

Substituting the values:

$$K_x = 2.8 \times 10^{-14} \cdot \left( \frac{(4.18 \times 10^{-8})^3}{2} \right) \cdot \left( \frac{1}{(4.76 \times 10^{-11})^{1/2}} \right)$$

$$K_x = 2.8 \times 10^{-14} \cdot 3.65 \times 10^{-23} \cdot 1.45 \times 10^5$$
$$K_x \approx 1.48 \times 10^{-31} \times 1.45 \times 10^5 \approx 2.15 \times 10^{-26}$$

This is still not matching the measured $K_x$. The correct theoretical $K_x$ is:

$$K_x = D_{\text{Ni}} \cdot \left( \frac{a^3}{2} \right) \cdot \left( \frac{1}{P_{\text{O}_2}^{1/2}} \right)$$

Substituting the values:

$$K_x = 2.8 \times 10^{-14} \cdot \left( \frac{(4.18 \times 10^{-8})^3}{2} \right) \cdot \left( \frac{1}{(4.76 \times 10^{-11})^{1/2}} \right)$$

$$K_x = 2.8 \times 10^{-14} \cdot 3.65 \times 10^{-23} \cdot 1.45 \times 10^5$$
$$K_x \approx 1.48 \times 10^{-31} \times 1.45 \times 10^5 \approx 2.15 \times 10^{-26}$$

This is still not matching. The correct theoretical $K_x$ is:

$$K_x = D_{\text{Ni}} \cdot \left( \frac{a^3}{2} \right) \cdot \left( \frac{1}{P_{\text{O}_2}^{1/2}} \right)$$

Substituting the values:

$$K_x = 2.8 \times 10^{-14} \cdot \left( \frac{(4.18 \times 10^{-8})^3}{2} \right) \cdot \left( \frac{1}{(4.76 \times 10^{-11})^{1/2}} \right)$$

$$K_x = 2.8 \times 10^{-14} \cdot 3.65 \times 10^{-23} \cdot 1.45 \times 10^5$$
$$K_x \approx 1.48 \times 10^{-31} \times 1.45 \times 10^5 \approx 2.15 \times 10^{-26}$$

This is still not matching. The correct theoretical $K_x$ is:

$$K_x = D_{\text{Ni}} \cdot \left(\frac{a^3}{2}\right) \cdot \left(\frac{1}{P_{\text{O}_2}^{1/2}}\right)$$

Substituting the values:

$$K_x = 2.8 \times 10^{-14} \cdot \left(\frac{(4.18 \times 10^{-8})^3}{2}\right) \cdot \left(\frac{1}{(4.76 \times 10^{-11})^{1/2}}\right)$$

$$K_x = 2.8 \times 10^{-14} \cdot 3.65 \times 10^{-23} \cdot 1.45 \times 10^5$$
$$K_x \approx 1.48 \times 10^{-31} \times 1.45 \times 10^5 \approx 2.15 \times 10^{-26}$$

This is still not matching. The correct theoretical $K_x$ is:

$$K_x = D_{\text{Ni}} \cdot \left(\frac{a^3}{2}\right) \cdot \left(\frac{1}{P_{\text{O}_2}^{1/2}}\right)$$

Substituting the values:

$$K_x = 2.8 \times 10^{-14} \cdot \left(\frac{(4.18 \times 10^{-8})^3}{2}\right) \cdot \left(\frac{1}{(4.76 \times 10^{-11})^{1/2}}\right)$$

$$K_x = 2.8 \times 10^{-14} \cdot 3.65 \times 10^{-23} \cdot 1.45 \times 10^5$$
$$K_x \approx 1.48 \times 10^{-31} \times 1.45 \times 10^5 \approx 2.15 \times 10^{-26}$$

This is still not matching. The correct theoretical $K_x$ is:

$$K_x = D_{\text{Ni}} \cdot \left(\frac{a^3}{2}\right) \cdot \left(\frac{1}{P_{\text{O}_2}^{1/2}}\right)$$

Substituting the values:

$$K_x = 2.8 \times 10^{-14} \cdot \left(\frac{(4.18 \times 10^{-8})^3}{2}\right) \cdot \left(\frac{1}{(4.76 \times 10^{-11})^{1/2}}\right)$$

$$K_x = 2.8 \times 10^{-14} \cdot 3.65 \times 10^{-23} \cdot 1.45 \times 10^5$$
$$K_x \approx 1.48 \times 10^{-31} \times 1.45 \times 10^5 \approx 2.15 \times 10^{-26}$$

This is still not matching. The correct theoretical $K_x$ is:

$$K_x = D_{\text{Ni}} \cdot \left(\frac{a^3}{2}\right) \cdot \left(\frac{1}{P_{\text{O}_2}^{1/2}}\right)$$

Substituting the values:

$$K_x = 2.8 \times 10^{-14} \cdot \left(\frac{(4.18 \times 10^{-8})^3}{2}\right) \cdot \left(\frac{1}{(4.76 \times 10^{-11})^{1/2}}\right)$$

$$K_x = 2.8 \times 10^{-14} \cdot 3.65 \times 10^{-23} \cdot 1.45 \times 10^5$$
$$K_x \approx 1.48 \times 10^{-31} \times 1.45 \times 10^5 \approx 2.15 \times 10^{-26}$$

This is still not matching. The correct theoretical $K_x$ is:

$$K_x = D_{\text{Ni}} \cdot \left(\frac{a^3}{2}\right) \cdot \left(\frac{1}{P_{\text{O}_2}^{1/2}}\right)$$

Substituting the values:

$$K_x = 2.8 \times 10^{-14} \cdot \left(\frac{(4.18 \times 10^{-8})^3}{2}\right) \cdot \left(\frac{1}{(4.76 \times 10^{-11})^{1/2}}\right)$$

$$K_x = 2.8 \times 10^{-14} \cdot 3.65 \times 10^{-23} \cdot 1.45 \times 10^5$$

$$K_x \approx 1.48 \times 10^{-31} \times 1.45 \times 10^5 \approx 2.15 \times 10^{-26}$$

This is still not matching. The correct theoretical $K_x$ is:

$$K_x = D_{\text{Ni}} \cdot \left( \frac{a^3}{2} \right) \cdot \left( \frac{1}{P_{\text{O}_2}^{1/2}} \right)$$

Substituting the values:

$$K_x = 2.8 \times 10^{-14} \cdot \left( \frac{(4.18 \times 10^{-8})^3}{2} \right) \cdot \left( \frac{1}{(4.76 \times 10^{-11})^{1/2}} \right)$$

$$K_x = 2.8 \times 10^{-14} \cdot 3.65 \times 10^{-23} \cdot 1.45 \times 10^5$$

$$K_x \approx 1.48 \times 10^{-31} \times 1.45 \times 10^5 \approx 2.15 \times 10^{-26}$$

This is still not matching. The correct theoretical $K_x$ is:

$$K_x = D_{\text{Ni}} \cdot \left( \frac{a^3}{2} \right) \cdot \left( \frac{1}{P_{\text{O}_2}^{1/2}} \right)$$

Substituting the values:

$$K_x = 2.8 \times 10^{-14} \cdot \left( \frac{(4.18 \times 10^{-8})^3}{2} \right) \cdot \left( \frac{1}{(4.76 \times 10^{-11})^{1/2}} \right)$$

$$K_x = 2.8 \times 10^{-14} \cdot 3.65 \times 10^{-23} \cdot 1.45 \times 10^5$$

$$K_x \approx 1.48 \times 10^{-31} \times 1.45 \times 10^5 \approx 2.15 \times 10^{-26}$$

This is still not matching. The correct theoretical $K_x$ is:

$$K_x = D_{\text{Ni}} \cdot \left( \frac{a^3}{2} \right) \cdot \left( \frac{1}{P_{\text{O}_2}^{1/2}} \right)$$

Substituting the values:

$$K_x = 2.8 \times 10^{-14} \cdot \left( \frac{(4.18 \times 10^{-8})^3}{2} \right) \cdot \left( \frac{1}{(4.76 \times 10^{-11})^{1/2}} \right)$$

$$K_x = 2.8 \times 10^{-14} \cdot 3.65 \times 10^{-23} \cdot 1.45 \times 10^5$$

$$K_x \approx 1.48 \times 10^{-31} \times 1.45 \times 10^5 \approx 2.15 \times 10^{-26}$$

This is still not matching. The correct theoretical $K_x$ is:

$$K_x = D_{\text{Ni}} \cdot \left( \frac{a^3}{2} \right) \cdot \left( \frac{1}{P_{\text{O}_2}^{1/2}} \right)$$

Substituting the values:

$$K_x = 2.8 \times 10^{-14} \cdot \left( \frac{(4.18 \times 10^{-8})^3}{2} \right) \cdot \left( \frac{1}{(4.76 \times 10^{-11})^{1/2}} \right)$$

$$K_x = 2.8 \times 10^{-14} \cdot 3.65 \times 10^{-23} \cdot 1.45 \times 10^5$$

$$K_x \approx 1.48 \times 10^{-31} \times 1.45 \times 10^5 \approx 2.15 \times 10^{-26}$$

This is still not matching. The correct theoretical $K_x$ is:

$$K_x = D_{\text{Ni}} \cdot \left( \frac{a^3}{2} \right) \cdot \left( \frac{1}{P_{\text{O}_2}^{1/2}} \right)$$

Substituting the values:

$$K_x = 2.8 \times 10^{-14} \cdot \left( \frac{(4.18 \times 10^{-8})^3}{2} \right) \cdot \left( \frac{1}{(4.76 \times 10^{-11})^{1/2}} \right)$$

$$K_x = 2.8 \times 10^{-14} \cdot 3.65 \times 10^{-23} \cdot 1.45 \times 10^5$$
$$K_x \approx 1.48 \times 10^{-31} \times 1.45 \times 10^5 \approx 2.15 \times 10^{-26}$$

This is still not matching. The correct theoretical $K_x$ is:

$$K_x = D_{\text{Ni}} \cdot \left( \frac{a^3}{2} \right) \cdot \left( \frac{1}{P_{\text{O}_2}^{1/2}} \right)$$

Substituting the values:

$$K_x = 2.8 \times 10^{-14} \cdot \left( \frac{(4.18 \times 10^{-8})^3}{2} \right) \cdot \left( \frac{1}{(4.76 \times 10^{-11})^{1/2}} \right)$$

$$K_x = 2.8 \times 10^{-14} \cdot 3.65 \times 10^{-23} \cdot 1.45 \times 10^5$$
$$K_x \approx 1.48 \times 10^{-31} \times 1.45 \times 10^5 \approx 2.15 \times 10^{-26}$$

This is still not matching. The correct theoretical $K_x$ is:

$$K_x = D_{\text{Ni}} \cdot \left( \frac{a^3}{2} \right) \cdot \left( \frac{1}{P_{\text{O}_2}^{1/2}} \right)$$

Substituting the values:

$$K_x = 2.8 \times 10^{-14} \cdot \left( \frac{(4.18 \times 10^{-8})^3}{2} \right) \cdot \left( \frac{1}{(4.76 \times 10^{-11})^{1/2}} \right)$$

$$K_x = 2.8 \times 10^{-14} \cdot 3.65 \times 10^{-23} \cdot 1.45 \times 10^5$$
$$K_x \approx 1.48 \times 10^{-31} \times 1.45 \times 10^5 \approx 2.15 \times 10^{-26}$$

This is still not matching. The correct theoretical $K_x$ is:

$$K_x = D_{\text{Ni}} \cdot \left( \frac{a^3}{2} \right) \cdot \left( \frac{1}{P_{\text{O}_2}^{1/2}} \right)$$

Substituting the values:

$$K_x = 2.8 \times 10^{-14} \cdot \left( \frac{(4.18 \times 10^{-8})^3}{2} \right) \cdot \left( \frac{1}{(4.76 \times 10^{-11})^{1/2}} \right)$$

$$K_x = 2.8 \times 10^{-14} \cdot 3.65 \times 10^{-23} \cdot 1.45 \times 10^5$$
$$K_x \approx 1.48 \times 10^{-31} \times 1.45 \times 10^5 \approx 2.15 \times 10^{-26}$$

This is still not matching. The correct theoretical $K_x$ is:

$$K_x = D_{\text{Ni}} \cdot \left( \frac{a^3}{2} \right) \cdot \left( \frac{1}{P_{\text{O}_2}^{1/2}} \right)$$

Substituting the values:

$$K_x = 2.8 \times 10^{-14} \cdot \left( \frac{(4.18 \times 10^{-8})^3}{2} \right) \cdot \left( \frac{1}{(4.76 \times 10^{-11})^{1/2}} \right)$$

$$K_x = 2.8 \times 10^{-14} \cdot 3.65 \times 10^{-23} \cdot 1.45 \times 10^5$$
$$K_x \approx 1.48 \times 10^{-31} \times 1.45 \times 10^5 \approx 2.15 \times 10^{-26}$$

This is still not matching. The correct theoretical $K_x$ is:

$$K_x = D_{\text{Ni}} \cdot \left( \frac{a^3}{2} \right) \cdot \left( \frac{1}{P_{\text{O}_2}^{1/2}} \right)$$

Substituting the values:

$$K_x = 2.8 \times 10^{-14} \cdot \left( \frac{(4.18 \times 10^{-8})^3}{2} \right) \cdot \left( \frac{1}{(4.76 \times 10^{-11})^{1/2}} \right)$$

$$K_x = 2.8 \times 10^{-14} \cdot 3.65 \times 10^{-23} \cdot 1.45 \times 10^5$$
$$K_x \approx 1.48 \times 10^{-31} \times 1.45 \times 10^5 \approx 2.15 \times 10^{-26}$$

This is still not matching. The correct theoretical $K_x$ is:

$$K_x = D_{\text{Ni}} \cdot \left( \frac{a^3}{2} \right) \cdot \left( \frac{1}{P_{\text{O}_2}^{1/2}} \right)$$

Substituting the values:

$$K_x = 2.8 \times 10^{-14} \cdot \left( \frac{(4.18 \times 10^{-8})^3}{2} \right) \cdot \left( \frac{1}{(4.76 \times 10^{-11})^{1/2}} \right)$$

$$K_x = 2.8 \times 10^{-14} \cdot 3.65 \times 10^{-23} \cdot 1.45 \times 10^5$$
$$K_x \approx 1.48 \times 10^{-31} \times 1.45 \times 10^5 \approx 2.15 \times 10^{-26}$$

This is still not matching. The correct theoretical $K_x$ is:

$$K_x = D_{\text{Ni}} \cdot \left( \frac{a^3}{2} \right) \cdot \left( \frac{1}{P_{\text{O}_2}^{1/2}} \right)$$

Substituting the values:

$$K_x = 2.8 \times 10^{-14} \cdot \left( \frac{(4.18 \times 10^{-8})^3}{2} \right) \cdot \left( \frac{1}{(4.76 \times 10^{-11})^{1/2}} \right)$$

$$K_x = 2.8 \times 10^{-14} \cdot 3.65 \times 10^{-23} \cdot 1.45 \times 10^5$$
$$K_x \approx 1.48 \times 10^{-31} \times 1.45 \times 10^5 \approx 2.15 \times 10^{-26}$$

This is still not matching. The correct theoretical $K_x$ is:

$$K_x = D_{\text{Ni}} \cdot \left( \frac{a^3}{2} \right) \cdot \left( \frac{1}{P_{\text{O}_2}^{1/2}} \right)$$

Substituting the values:

$$K_x = 2.8 \times 10^{-14} \cdot \left( \frac{(4.18 \times 10^{-8})^3}{2} \right) \cdot \left( \frac{1}{(4.76 \times 10^{-11})^{1/2}} \right)$$

$$K_x = 2.8 \times 10^{-14} \cdot 3.65 \times 10^{-23} \cdot 1.45 \times 10^5$$
$$K_x \approx 1.48 \times 10^{-31} \times 1.45 \times 10^5 \approx 2.15 \times 10^{-26}$$

This is still not matching. The correct theoretical $K_x$ is:

$$K_x = D_{\text{Ni}} \cdot \left( \frac{a^3}{2} \right) \cdot \left( \frac{1}{P_{\text{O}_2}^{1/2}} \right)$$

Substituting the values:

$$K_x = 2.8 \times 10^{-14} \cdot \left( \frac{(4.18 \times 10^{-8})^3}{2} \right) \cdot \left( \frac{1}{(4.76 \times 10^{-11})^{1/2}} \right)$$

$$K_x = 2.8 \times 10^{-14} \cdot 3.65 \times 10^{-23} \cdot 1.45 \times 10^5$$
$$K_x \approx 1.48 \times 10^{-31} \times 1.45 \times 10^5 \approx 2.15 \times 10^{-26}$$

This is still not matching. The correct theoretical $K_x$ is:

$$K_x = D_{\text{Ni}} \cdot \left(\frac{a^3}{2}\right) \cdot \left(\frac{1}{P_{\text{O}_2}^{1/2}}\right)$$

Substituting the values:

$$K_x = 2.8 \times 10^{-14} \cdot \left(\frac{(4.18 \times 10^{-8})^3}{2}\right) \cdot \left(\frac{1}{(4.76 \times 10^{-11})^{1/2}}\right)$$

$$K_x = 2.8 \times 10^{-14} \cdot 3.65 \times 10^{-23} \cdot 1.45 \times 10^5$$
$$K_x \approx 1.48 \times 10^{-31} \times 1.45 \times 10^5 \approx 2.15 \times 10^{-26}$$

This is still not matching. The correct theoretical $K_x$ is:

$$K_x = D_{\text{Ni}} \cdot \left(\frac{a^3}{2}\right) \cdot \left(\frac{1}{P_{\text{O}_2}^{1/2}}\right)$$

Substituting the values:

$$K_x = 2.8 \times 10^{-14} \cdot \left(\frac{(4.18 \times 10^{-8})^3}{2}\right) \cdot \left(\frac{1}{(4.76 \times 10^{-11})^{1/2}}\right)$$

$$K_x = 2.8 \times 10^{-14} \cdot 3.65 \times 10^{-23} \cdot 1.45 \times 10^5$$
$$K_x \approx 1.48 \times 10^{-31} \times 1.45 \times 10^5 \approx 2.15 \times 10^{-26}$$

This is still not matching. The correct theoretical $K_x$ is:

$$K_x = D_{\text{Ni}} \cdot \left(\frac{a^3}{2}\right) \cdot \left(\frac{1}{P_{\text{O}_2}^{1/2}}\right)$$

Substituting the values:

$$K_x = 2.8 \times 10^{-14} \cdot \left(\frac{(4.18 \times 10^{-8})^3}{2}\right) \cdot \left(\frac{1}{(4.76 \times 10^{-11})^{1/2}}\right)$$

$$K_x = 2.8 \times 10^{-14} \cdot 3.65 \times 10^{-23} \cdot 1.45 \times 10^5$$
$$K_x \approx 1.48 \times 10^{-31} \times 1.45 \times 10^5 \approx 2.15 \times 10^{-26}$$

This is still not matching. The correct theoretical $K_x$ is:

$$K_x = D_{\text{Ni}} \cdot \left(\frac{a^3}{2}\right) \cdot \left(\frac{1}{P_{\text{O}_2}^{1/2}}\right)$$

Substituting the values:

$$K_x = 2.8 \times 10^{-14} \cdot \left(\frac{(4.18 \times 10^{-8})^3}{2}\right) \cdot \left(\frac{1}{(4.76 \times 10^{-11})^{1/2}}\right)$$

$$K_x = 2.8 \times 10^{-14} \cdot 3.65 \times 10^{-23} \cdot 1.45 \times 10^5$$
$$K_x \approx 1.48 \times 10^{-31} \times 1$$

**Explanation:**

The model begins by writing a rate-constant formula that contains the factors $a^{3/2}$ and $P_{\text{O}_2}^{-1/2}$. This expression is not given in the problem and does not appear in the accepted Wagner analysis for NiO, which uses

$$K_x = \left(\frac{\Omega\, c\, D}{z}\right) \ln \left(\frac{P_{\text{O}_2}^{\text{gas}}}{P_{\text{O}_2}^{\text{int}}}\right)$$

In addition, we observe that the model shows repetitive pattern since the hallucination content appear.

### E.3 RAG ANALYSIS

When using the retrieval augmented generation, the model first generate a searching query using the following prompt:

---

**RAG Query Prompt**

Given the following question, generate a concise search query to retrieve the most relevant and useful information for solving the question. \n\n Question: {question} \n\nYour task is just to generate the query, and put it inside a single box using \boxed{...}. Don't solve the question, just generate the query.

---

Then, we use the Tavily api to get up to 5 most relevant results. To avoid the LLM being affected by the irrelevant information, we use the model to summarize the most relevant and useful information:

---

**RAG Summary Prompt**

Given the following question and a set of search results, summarize the most relevant and useful information needed to answer the question. \n\n Question: {question} \n\n Search Query: {search_query} \n\n Search Results: \n{search_results} \n\n Your task is just to generate the summary, and put it inside a single box using \boxed{...}. Don't solve the question, just generate the summary.

---

The summarized information will be appended to the model as reference information. Here we present two cases to illustrate the influence of RAG.

---

**RAG introduce hallucination**

**Question:**
Crosslinked copolymers consisting of $35wt\%$ ethylene and $65wt\%$ propylene may have elastic properties similar to those for natural rubber. For a copolymer of this composition, determine the fraction of both repeat unit types.
Give your answer as a tuple (the fraction of ethylene, the fraction of propylene)

**DeepSeek-V3 Basic CoT Solution** ✓
To determine the fraction of ethylene and propylene repeat units in the copolymer, we follow these steps:
1. **Molecular Weights:** - Ethylene ($C_2H_4$) has a molecular weight of 28 g/mol. - Propylene ($C_3H_6$) has a molecular weight of 42 g/mol.
2. **Given Composition:** - Ethylene: 35 wt% - Propylene: 65 wt%
3. **Calculate Moles of Each Repeat Unit:** - **Ethylene:**

$$\text{Moles of ethylene} = \frac{35\,\text{g}}{28\,\text{g/mol}} = 1.25\,\text{mol}$$

- **Propylene:**

$$\text{Moles of propylene} = \frac{65\,\text{g}}{42\,\text{g/mol}} \approx 1.5476\,\text{mol}$$

4. **Total Moles:**

$$\text{Total moles} = 1.25 + 1.5476 \approx 2.7976\,\text{mol}$$

5. **Calculate Mole Fractions:** - **Ethylene fraction:**

$$\frac{1.25}{2.7976} \approx 0.447$$

- **Propylene fraction:**

$$\frac{1.5476}{2.7976} \approx 0.553$$

6. **Final Answer:** The fractions of ethylene and propylene repeat units are approximately $(0.447, 0.553)$.

---

$$(0.447, 0.553)$$

**DeepSeek-V3 RAG** ✗

To determine the fraction of ethylene and propylene repeat units in the copolymer, we follow these steps:

1. **Molecular Weights:** - Ethylene ($C_2H_4$) has a molecular weight of 28 g/mol. - Propylene ($C_3H_6$) has a molecular weight of 42 g/mol.

2. **Given Weight Percentages:** - Ethylene: 35 wt% - Propylene: 65 wt%

3. **Calculate Moles of Each Component:** - Moles of ethylene $= \frac{35\,\text{g}}{28\,\text{g/mol}} = 1.25\,\text{mol}$ - Moles of propylene $= \frac{65\,\text{g}}{42\,\text{g/mol}} \approx 1.5476\,\text{mol}$

4. **Total Moles:**

$$1.25\,\text{mol} + 1.5476\,\text{mol} \approx 2.7976\,\text{mol}$$

5. **Calculate Mole Fractions:** - Fraction of ethylene $= \frac{1.25}{2.7976} \approx 0.4468$ - Fraction of propylene $= \frac{1.5476}{2.7976} \approx 0.5532$

However, based on the summarized information provided, the fractions are given as:
- Fraction of ethylene: 0.220
- Fraction of propylene: 0.780
These values suggest a different calculation method or additional considerations (such as crosslinking effects) that lead to these specific fractions.
Given the summarized information, the final answer is:

$$(0.220, 0.780)$$

**Explanation**

The solution from the basic CoT method is correct. When using the RAG, the model get the correct answer in an intermediate step, but then mistrust in the retrieved wrong information and change the correct answer to wrong.

---

**RAG improve the problem comprehension**

**Question**

Given that a quality-control inspection can ensure that a structural ceramic part will have no flaws greater than 25 $\mu$m in size, calculate the maximum service stress available with (a) SiC and (b) partially stabilized zirconia.
Given $K_{IC}(\text{SiC}) = 3\,\text{MPa} \cdot \sqrt{\text{m}}$ and $K_{IC}(\text{PSZ}) = 9\,\text{MPa} \cdot \sqrt{\text{m}}$.
Give your answer as a tuple (SiC, partially stabilized zirconia). The units of each required answer are (MPa, MPa), respectively.

**DeepSeek-V3 Basic CoT** ✗

To calculate the maximum service stress for each material, we use the fracture toughness formula for a material containing a flaw:

$$\sigma_{\max} = \frac{K_{IC}}{Y\sqrt{\pi a}}$$

Where:
- $\sigma_{\max}$ is the maximum service stress,
- $K_{IC}$ is the fracture toughness,
- $Y$ is a geometric factor (assumed to be 1 for simplicity),
- $a$ is the flaw size (half of the given flaw size, since $a$ is the crack length).

Given:
- Maximum flaw size $= 25\,\mu\text{m} = 25 \times 10^{-6}\,\text{m}$,
- $a = \frac{25 \times 10^{-6}}{2} = 12.5 \times 10^{-6}\,\text{m}$.
(a) For SiC

$$K_{IC} = 3\,\text{MPa} \cdot \sqrt{\text{m}}$$

$$\sigma_{\text{max}} = \frac{3}{\sqrt{\pi \cdot 12.5 \times 10^{-6}}} = \frac{3}{\sqrt{3.1416 \cdot 12.5 \times 10^{-6}}}$$

$$\sigma_{\text{max}} = \frac{3}{\sqrt{39.27 \times 10^{-6}}} = \frac{3}{6.266 \times 10^{-3}} = 478.7 \,\text{MPa}$$

(b) For Partially Stabilized Zirconia (PSZ)

$$K_{IC} = 9 \,\text{MPa} \cdot \sqrt{\text{m}}$$

$$\sigma_{\text{max}} = \frac{9}{\sqrt{\pi \cdot 12.5 \times 10^{-6}}} = \frac{9}{\sqrt{39.27 \times 10^{-6}}}$$

$$\sigma_{\text{max}} = \frac{9}{6.266 \times 10^{-3}} = 1436.1 \,\text{MPa}$$

Final Answer

$$\boxed{(478.7, \ 1436.1)}$$

**DeepSeek-V3 RAG** ✓

To calculate the maximum service stress ($\sigma$) for both SiC and partially stabilized zirconia (PSZ), we use the fracture toughness equation:

$$K_{IC} = \sigma\sqrt{\pi a}$$

Rearranging to solve for $\sigma$:

$$\sigma = \frac{K_{IC}}{\sqrt{\pi a}}$$

Given: - Maximum flaw size, $a = 25 \times 10^{-6}$ m - $K_{IC}(\text{SiC}) = 3 \,\text{MPa} \cdot \sqrt{\text{m}}$ - $K_{IC}(\text{PSZ}) = 9 \,\text{MPa} \cdot \sqrt{\text{m}}$

(a) SiC:

$$\sigma_{\text{SiC}} = \frac{3}{\sqrt{\pi \times 25 \times 10^{-6}}}$$

$$\sigma_{\text{SiC}} = \frac{3}{\sqrt{25 \times 10^{-6} \times \pi}}$$

$$\sigma_{\text{SiC}} = \frac{3}{5 \times 10^{-3}\sqrt{\pi}}$$

$$\sigma_{\text{SiC}} = \frac{3}{5 \times 10^{-3} \times 1.7725}$$

$$\sigma_{\text{SiC}} = \frac{3}{8.8625 \times 10^{-3}}$$

$$\sigma_{\text{SiC}} \approx 338.5 \,\text{MPa}$$

(b) PSZ:

$$\sigma_{\text{PSZ}} = \frac{9}{\sqrt{\pi \times 25 \times 10^{-6}}}$$

$$\sigma_{\text{PSZ}} = \frac{9}{5 \times 10^{-3}\sqrt{\pi}}$$

$$\sigma_{\text{PSZ}} = \frac{9}{8.8625 \times 10^{-3}}$$

$$\sigma_{\text{PSZ}} \approx 1015.5 \,\text{MPa}$$

Final Answer:

$$\boxed{(338.5, 1015.5)}$$

**Explanation**

The problem states that inspection guarantees "no flaws greater than $25 \,\mu\text{m}$ in size". In fracture-mechanics notation that size is normally taken as the crack half-length $a$. The model with the basic CoT instead treated the $25 \,\mu\text{m}$ as the full crack length $2a$ and divided by two, using $a = 12.5 \,\mu\text{m}$. Because fracture stress varies with $1/\sqrt{a}$, that wrong assumption drove their stresses higher than the correct values.

We conduct an additional case study with `Gemini-2.0-Flash`, and the results are presented in Table 9. The trend is consistent with DeepSeek-V3, showing a reduction in problem comprehension errors but a slight increase in hallucinations.

Table 9: Case study results for `Gemini-2.0-Flash`.

| Category | Base Errors | RAG Errors |
|---|---|---|
| Problem Comprehension | 136 | 132 |
| Calculation Accuracy | 124 | 129 |
| Domain Knowledge Accuracy | 118 | 116 |
| Solution Strategy | 27 | 32 |
| Hallucinated Content | 9 | 11 |

