# OpenReview forum: "MatSciBench: Benchmarking the Reasoning Ability of LLM in Materials Science"
_ICLR.cc/2026/Conference — ICLR 2026 Conference Desk Rejected Submission_

### Official Review · Reviewer_vfWw · 2025-10-29

**Soundness:** 3
**Presentation:** 2
**Contribution:** 3
**Rating:** 6
**Confidence:** 4

**Summary:**

This paper presents a new benchmark, MatSciBench, which is specifically designed for the subject of material science. The benchmark consists of 1340 questions, extracted from undergraduate and graduate textbook.
The questions are further curated by human experts for better quality. Some questions involve images, which means requiring multimodal reasoning ability.
The paper benchmark the performance of latest LLMs with different reasoning paradigms.

**Strengths:**

- A carefully curated benchmark is a good contribution to the community, especially for the under-explored domain of material science.

- The benchmark contains multimodal questions.

- Good data size.

- Good analysis on the stats of the dataset.

**Weaknesses:**

- More information needed for:
  - Line 175, the parsing algorithm of identifying the problems and solutions.
  - the details on background of Human experts who reviewed the extracted questions. Is there any cross validation?

**Questions:**

Can you have a paragraph elaborating how the questions in material science different from other domains, such as math, physics and chemistry.

---

> ### Author Response · Authors · 2025-12-01
> **Response to Reviewer vfWw**
>
> We thank the reviewer for their constructive feedback and for recognizing the value of MatSciBench as a comprehensive, multimodal benchmark for the under-explored domain of materials science. We appreciate the positive assessment of our dataset size and analysis. Below, we address the specific weaknesses and questions raised.
>
> ### W1: The parsing algorithm of identifying the problems and solutions.
> We will updated the manuscript to provide a more detailed description of our multi-stage extraction process:
> - Digitization: We first utilize Mistral OCR to convert the raw PDF textbook content into text, preserving the layout information.
> - Candidate Identification (Regex): We employ regular expressions to locate problem anchors (e.g., keywords like "Question," "Exercise," "Practice," "Solution").
> - Context Extraction: Upon locating an anchor, we extract the subsequent text window (approximately 100 lines) to capture the full context of the problem statement and its corresponding solution.
> - LLM-Assisted Parsing: An LLM is then used to structure this raw text into our standardized json format.
> - Human Verification: Crucially, this automated pipeline is followed by a manual review. Every extracted question-answer pair is verified against the original source to ensure the OCR accuracy, the correctness of the separation between problem and solution, and the integrity of any associated mathematical formulas.
>
> ### W2: The background of the human expert.
> The human expert supervising the dataset preparing is a material science postdoc.
>
> ### Q1: How the questions in material science differ from other domains?
> This is an excellent suggestion. We will add a dedicated discussion to the manuscript to clarify this distinction. Materials science is not simply a subset of physics or chemistry but a unique discipline that bridges fundamental science and engineering applications.
> - Interdisciplinary Integration: Unlike pure physics or chemistry problems which often focus on isolating a single principle, materials science questions inherently require knowledge integration. A single problem often demands reasoning across thermodynamics (physics), bonding (chemistry), and mechanical behavior (engineering) to predict a material's performance.
> - Specific Taxonomy: As reflected in our benchmark’s taxonomy, materials science involves unique categories such as Structures (e.g., micro/nano-structure, crystal defects) and Processing (e.g., synthesis, heat treatment). These concepts require reasoning about how processing alters structure, which in turn determines properties.
> - Multimodality: Materials science is highly visual. Reasoning often requires interpreting specialized visual data like phase diagrams, micrographs, and crystal structure unit cells. This requires a multimodal reasoning capability distinct from the figures often found in other domains.

---

### Official Review · Reviewer_nUjU · 2025-10-31

**Soundness:** 2
**Presentation:** 2
**Contribution:** 2
**Rating:** 4
**Confidence:** 2

**Summary:**

The paper introduces **MatSciBench**, a new benchmark designed to evaluate the reasoning capabilities of Large Language Models (LLMs) in the domain of materials science. The benchmark consists of 1,340 college-level problems sourced from 10 textbooks, covering 6 primary fields and 31 sub-fields of materials science. Key features of MatSciBench include a three-tier difficulty classification based on reasoning length, detailed reference solutions for error analysis, and the inclusion of 315 multimodal questions with visual contexts. The authors conduct an extensive evaluation of several "thinking" and "non-thinking" models, analyzing the efficacy of different reasoning strategies like Chain-of-Thought (CoT), tool-augmentation, and self-correction. The primary finding is that even the best-performing model, "Gemini-2.5-Pro," scores below 80%, indicating the benchmark's difficulty.

**Strengths:**

*   **Significance:** The work addresses a clear and important gap. While numerous STEM benchmarks exist, materials science—a critical, interdisciplinary field at the intersection of physics, chemistry, and engineering—has been underexplored. Creating a dedicated, high-quality benchmark for this domain is a significant step towards understanding and improving LLM capabilities for real-world scientific applications.

*   **Quality:**
    *   **Data Sourcing:** The problems are curated from established college-level textbooks, not synthetically generated by LLMs. This ensures the questions are authentic, relevant, and representative of the knowledge required in the field.
    *   **Rigorous Curation:** The inclusion of expert review, detailed reference solutions for over 900 questions, and a structured, hierarchical taxonomy demonstrates a high level of rigor.

*   **Originality:** While the concept of a benchmark is not new, MatSciBench is original in its specific focus, depth, and quality for the materials science domain.

**Weaknesses:**

*   **Limited Scope of RAG Case Study:** The RAG analysis is presented as a "case study" but is too limited to support generalizable conclusions. It primarily uses one model (DeepSeek-V3) and one RAG implementation (general web search). The finding that RAG improves comprehension but not knowledge accuracy is fascinating and counter-intuitive, but it could be an artifact of the specific model or the noisy nature of web search for specialized topics. A more robust analysis would involve multiple models and, crucially, a domain-specific retrieval corpus (e.g., the source textbooks themselves).

*   **Difficulty Classification Metric:** Using response length as a proxy for difficulty is a pragmatic and clever approach, and the authors do well to validate it by showing a negative correlation with accuracy. However, this metric could be noisy. For some models, increased length might simply reflect higher verbosity or a tendency to "show work" on all problems, not necessarily an increase in reasoning complexity. A small-scale validation against human expert ratings of difficulty would have made this classification unassailable.

**Questions:**

1.  **Regarding the RAG Analysis:**
    *   The finding that RAG can increase hallucination is important. Do you believe this is a general risk, or is it specific to using a broad web search for highly technical queries where precision is low?
    *   Have you considered conducting a more controlled RAG experiment where the retrieval corpus is composed of the digitized textbooks from which the questions were sourced? This would test the model's ability to "use a reference" more directly and might yield different results regarding knowledge-based errors.

2.  **Regarding Multimodal Reasoning:**
    *   The analysis identifies spatial reasoning and precise data extraction as key challenges. Could you include one or two specific examples in the appendix showing a model failing to interpret a crystal lattice diagram or misreading values from a phase diagram? This would powerfully illustrate the nature of the current limitations.

---

> ### Author Response · Authors · 2025-12-01
> **Response to Reviewer nUjU**
>
> We thank the reviewer for recognizing the significance of MatSciBench and the rigor of our data curation. We value the constructive feedback regarding our RAG analysis and difficulty metrics.
>
> ### W1 & Q1: Scope of RAG Case Study & Hallucination Risks
> We agree that a domain-specific retrieval corpus (e.g., textbooks) is the gold standard for accuracy. However, our current "Web Search" RAG setup was designed to simulate the most common user behavior: using general-purpose search to solve specialized problems. To address your concerns about generalization:
> - New Experiment: We have extended the RAG analysis to Gemini-2.0-Flash (see Appendix E.3). The results align with our DeepSeek-V3 findings: while retrieval improves context, it often introduces noise in highly technical domains, leading to "confident hallucinations."
> - Web vs. Textbook: We believe the risk of increased hallucination is heightened by the noise of broad web search. We will clarify in the paper that our current findings represent "Open-Web RAG" performance.
> - Revision Plan: While we cannot finish the full textbook-RAG experiment during the rebuttal window, we plan to include a comparison in the final version to explicitly contrast "Web RAG" vs. "Domain RAG" performance.
>
>
> ### W2: Difficulty Classification Metric Response
> We acknowledge that response length is a proxy. However, we believe it is a robust signal because it correlates strongly with the number of reasoning steps required for college-level derivations and shows different accuracy rates in different difficulty groups. To validate this, we will include a human validation study on a subset of problems in the final version. We will report the correlation between these expert ratings and our length-based metric to ensure the taxonomy is unassailable.
>
> ### Q2: Multimodal Reasoning Examples Response
> This is an excellent suggestion that highlights the specific failure modes of current MLLMs. We will add a dedicated subsection in the appendix showcasing these failures. For example, we have identified cases where:
> - Phase Diagrams and Curves: Models fail to read numbers correctly from the figure or fail to distinguish different lines when they are close.
> - 3D Figures: Models fail to perform spatial counting (e.g., counting atoms in a unit cell) when presented with 3D perspective diagrams, or the model fails to recognize the lattice type.

---

### Official Review · Reviewer_b6nC · 2025-11-01

**Soundness:** 3
**Presentation:** 2
**Contribution:** 3
**Rating:** 4
**Confidence:** 4

**Summary:**

This paper presents MatSciBench, a benchmark of 1,340 expert-curated, college-level materials science problems aimed at evaluating large language models' (LLMs) reasoning abilities in an underexplored yet crucial scientific domain. MatSciBench spans 6 primary fields and 31 sub-fields, classifies difficulty on a three-tier scale, and incorporates detailed reference solutions and over 300 multimodal (image-based) questions. The paper systematically evaluates a suite of SOTA LLMs—distinguishing between “thinking” and “non-thinking” models—across prompting strategies (chain-of-thought, tool-augmentation, self-correction), analyzes performance along multiple dimensions (difficulty, efficiency, multimodal, error type), and provides nuanced insights into LLMs’ scientific reasoning in materials science.

**Strengths:**

-  The benchmark offers one of the most comprehensive and fine-grained materials science evaluations, comprising 1,340 real questions derived from a wide spectrum of authoritative university-level textbooks.

- The inclusion of 315 image-based questions establishes a new standard for assessing multimodal reasoning in materials science, backed by dedicated quantitative analysis.

-  The experimental design is notably comprehensive, including extensive evaluation of SOTA LLMs across multiple prompting strategies and a detailed analysis of error types, which provides meaningful diagnostic insights.

**Weaknesses:**

- The motivation for this study requires further elaboration. Given the existence of numerous established scientific reasoning benchmarks (e.g., MMMU-PRO for multimodal reasoning, GPQA/SuperGPQA for graduate-level reasoning, SciEval, SciBench for scientific evaluation), the authors must clearly contrast and articulate the unique advantages and research necessity of MatSciBench against these established contemporaries.
- The evaluation framework is not fully documented. Specifically, the proportional distribution of questions across different data subcategories/subfields needs clearer articulation. Furthermore, the description of the evaluation metrics used is vague; a precise clarification of the scoring mechanism (e.g., per-question calculation and overall aggregation) is required for reproducibility.
- While the study provides diverse question categorizations, it lacks deep insights into performance disparities across different model types (i.e., how model families handle specific sub-fields). Moreover, the multimodal analysis is insufficient, notably lacking evaluations of powerful open-source Vision-Language (VL) models such as Qwen3-VL and InternVL-3.

**Questions:**

same as weakness

---

> ### Author Response · Authors · 2025-12-01
> **Response to Reviewer b6nC**
>
> We thank the reviewer for their constructive feedback and for recognizing MatSciBench as one of the "most comprehensive and fine-grained materials science evaluations" with a notable experimental design. We address your specific concerns below.
>
> ### W1: Motivation and Comparison with Existing Benchmarks
> While we agree that benchmarks like MMMU-Pro, SciBench, and SuperGPQA cover broad scientific reasoning, MatSciBench addresses critical gaps that these general benchmarks overlook:
> - Depth vs. Breadth: General benchmarks cover materials science only superficially as a sub-topic. In contrast, MatSciBench provides fine-grained granularity with 31 specific sub-fields (e.g., Crystal Defects, Phase Diagrams) derived from 10 authoritative textbooks, allowing for the diagnosis of specific domain weaknesses that general benchmarks cannot isolate.
> - Data Quality (Real vs. Synthetic): Unlike some recent materials-specific attempts that rely on synthetic, LLM-generated data, MatSciBench consists entirely of expert-curated, college-level problems, ensuring scientific validity and eliminating the "noise" inherent in synthetic benchmarks.
> - Domain-Specific Multimodality: While general benchmarks include images, they do not focus on the specialized visual literacy required in materials science (e.g., interpreting phase diagrams, crystal lattices, or microscopic structures). MatSciBench includes 315 such specialized multimodal questions to test these unique spatial and interpretative skills.
>
> ### W2: Documentation of Evaluation Framework and Distribution of Questions
> We will revise Section 3 and 4 to include more details.
>
> Distribution: The distribution of questions is explicitly categorized by our taxonomy. As detailed in Section 3.3 and visualized in Figure 2, the dataset spans 6 primary fields (Materials, Properties, Structures, etc.) and 31 sub-fields. The difficulty distribution is 50.7% Easy, 29.1% Medium, and 20.1% Hard.
>
> Scoring Mechanism: To ensure correct evaluation, we use a rule-based evaluation system combined with LLM-based judge to prevent bias. We first extract the answer from \box{...}, and then
> - Numerical Questions: check if it falls within a specific tolerance range (±5%) of the reference solution to account for rounding differences.
> - Formula Questions: We use the evaluation method in the Qwen-2.5-Math report [1].
>
> And if the LLM judge is enabled, we will use the LLM to decide whether the extracted answer is the same as the reference answer since the rule based method may fail to handle some edge cases.
> - Aggregation: Correct/wrong are decided per question and then aggregated to compute average accuracy per sub-field and difficulty level.
>
> ### W3: Insights and Multimodal Analysis
> We appreciate the suggestion to expand our model analysis.
> - New Models: In our revision, we will include evaluations of the requested open-source Vision-Language models, specifically Qwen3-VL and InternVL-3, to provide a more robust assessment of open-source multimodal capabilities.
> - Model Disparities: We will deepen our discussion on performance disparities. Our current analysis (Section 5) already highlights a key distinction between "thinking" models (e.g., Gemini-2.5-Pro, DeepSeek-R1) and "non-thinking" models. We observed that "thinking" models are significantly less sensitive to question difficulty and maintain performance on "Hard" questions better than non-thinking models (Section 5.1). We also discussed the failure case of different model families in Section 5.4.
>
> References:
>
> [1] https://github.com/QwenLM/Qwen2.5-Math

---

### Official Review · Reviewer_WJ8b · 2025-11-02

**Soundness:** 2
**Presentation:** 2
**Contribution:** 2
**Rating:** 4
**Confidence:** 4

**Summary:**

This paper introduces a well-designed benchmark to fill an important gap in evaluating LLMs in materials science. It offers strong dataset organization, rigorous evaluation, and demonstrates that current LLMs still lack deep scientific reasoning capacity in this domain.

**Strengths:**

1. This paper introduces first comprehensive benchmark focused specifically on materials science reasoning for LLMs.

2. This benchmark covers 6 fields and 31 subfields, with a well-organized taxonomy and difficulty levels.

3. This benchmark also contains visual reasoning tasks using diagrams, figures, and context images.

**Weaknesses:**

1. This benchmark lacks comparison with human performance, making it unclear how difficult the benchmark truly is or whether it reaches "college-level" difficulty.

2. The paper claims that RAG fails to reduce knowledge errors and may induce hallucinations. However, prior work[1] shows that when using domain-specific retrieval and query refinement, RAG can significantly improve factual accuracy. I suggest the authors soften this claim and clarify that the negative result stems from their specific RAG setup (general web search, no scientific databases or retrieval filtering), rather than RAG being ineffective in scientific domains overall.

3. This paper indicates a substantial drop in performance on image-based questions.  The decline could also stem from dataset design (e.g., 3D structural images, low-resolution diagrams), mismatched image-question alignment, or higher intrinsic difficulty of visual questions. The authors should conduct more detailed analysis on this section.

[1] Evaluating Retrieval-Augmented Generation Agents for Autonomous Scientific Discovery in Astrophysics.

**Questions:**

see weakness

---

> ### Author Response · Authors · 2025-12-01
> **Response to Reviewer WJ8b**
>
> We thank the reviewer for the constructive feedback and for recognizing MatSciBench as a comprehensive, well-organized benchmark with strong taxonomy design and visual reasoning capabilities. We address each concern below:
>
> ### W1: Comparison with human performance and "college-level" difficulty
> We appreciate this suggestion for strengthening the difficulty calibration. We claim the benchmark is "college-level" because:
> - Data source: All 1,340 questions are sourced exclusively from 10 widely-adopted undergraduate and graduate textbooks (Table 3, Appendix B.1), selected and validated by materials science experts.
> - Model performance ceiling: Even the best-performing model (Gemini-2.5-Pro) achieves only 77.37% on text-only questions, suggesting substantial headroom that aligns with challenging college-level material science knowledge.
>
> For the final version, we plan to add a human expert baseline on a representative subset (~100 questions across difficulty levels) to provide a concrete performance anchor. We will recruit materials science college students to establish this baseline.
>
> ### W2: Claims regarding RAG limitations
> We thank the reviewer for the insightful assessment and the reference [1]. We agree that our findings are specific to our experimental setup—"Open-Web RAG" via general search APIs—rather than optimized, domain-specific retrieval.
> To address this concern and clarify the scope of our claims:
> - Clarification of Scope: We will revise Section 5.5 to explicitly frame our results as limitations of "Open-Web RAG." We chose this setup to simulate the most common user behavior (using general-purpose tools to solve specialized problems), but we acknowledge that this differs from using curated scientific databases. We will cite [1] to highlight how specialized RAG can improve accuracy, contrasting it with our findings on general web search.
> - New Experiment (Appendix E.3): We have extended our RAG analysis to Gemini-2.0-Flash. The results align with our DeepSeek-V3 findings: while broad web retrieval provides context, it frequently introduces noise in highly technical domains, leading to  hallucinations. This confirms that the issue is common across models when using open web sources.
> - Future Addition: We plan to include a comparison in the final version contrasting "Web RAG" vs. "Domain RAG" (using textbooks) to empirically demonstrate the difference in factual accuracy.
>
> ### W3: Further analysis of the performance drop on image-based questions
> The performance drop stems from specific visual challenges inherent to materials science.
> In the final version, we will expand Section 5.3 to include:
>
> Breakdown by image type: Our dataset contains (a) phase diagrams and curves requiring understanding and precise value extraction, (b) 3D crystal structures demanding spatial reasoning. We will find failure cases separately for each category.
>
> Error analysis: Preliminary examination reveals the following cases:
>
>  - Phase diagrams and curves: Models fail to read numbers correctly from the figure or fail to distinguish different lines when they are close.
>
> - 3D figures: Models fail to perform spatial counting (e.g., counting atoms in a unit cell) when presented with 3D perspective diagrams, or the model fails to recognize the lattice type.
>
> We will provide examples in the final version.
>
>
> References
>
> [1] Evaluating Retrieval-Augmented Generation Agents for Autonomous Scientific Discovery in Astrophysics.

---

### Note · Program_Chairs · 2026-01-17
**Submission Desk Rejected by Program Chairs**

The following references in this submission do not refer to real documents and/or have major errors in bibliographic information:

 Daniel Reker, Tobias Gensch, Lenneard Hüsken, Alexey Nikolaev, Hong S. Li, Hans Stärk, Alex Zhavoronkov, and Alán Aspuru-Guzik. Autonomous discovery of small-molecule biologically active compounds. Nature Machine Intelligence, 5(6):650-661, 2023. doi: 10.1038/s42256-023-00669-7.

J. M. Rickman, T. Lookman, and S. V. Kalinin. Explainable machine learning in materials science. Nature Reviews Materials, 4:785-787, 2019. doi: 10.1038/s41578-019-0148-0.